# Homeostatic control of cell wall hydrolysis by the WalRK two-component signaling pathway in *Bacillus subtilis*

Genevieve S Dobihal[†], Yannick R Brunet[†], Josué Flores-Kim, David Z Rudner*

Department of Microbiology, Harvard Medical School, Boston, United States

**Abstract** Bacterial cells are encased in a peptidoglycan (PG) exoskeleton that protects them from osmotic lysis and specifies their distinct shapes. Cell wall hydrolases are required to enlarge this covalently closed macromolecule during growth, but how these autolytic enzymes are regulated remains poorly understood. *Bacillus subtilis* encodes two functionally redundant D,L-endopeptidases (CwlO and LytE) that cleave peptide crosslinks to allow expansion of the PG meshwork during growth. Here, we provide evidence that the essential and broadly conserved WalR-WalK two component regulatory system continuously monitors changes in the activity of these hydrolases by sensing the cleavage products generated by these enzymes and modulating their levels and activity in response. The WalR-WalK pathway is conserved among many Gram-positive pathogens where it controls transcription of distinct sets of PG hydrolases. Cell wall remodeling in these bacteria may be subject to homeostatic control mechanisms similar to the one reported here.

**\*For correspondence:**
rudner@hms.harvard.edu

[†]These authors contributed equally to this work

**Competing interests:** The authors declare that no competing interests exist.

## Introduction

The cell wall peptidoglycan (PG) is composed of long glycan strands cross-linked together by short peptides. This three-dimensional exoskeleton specifies shape and protects the cell from osmotic rupture. For cells to grow they must enlarge this covalently closed macromolecule and this requires both the synthesis of new material and hydrolysis of the PG meshwork to allow for its expansion. How the cell maintains the appropriate levels of these potentially autolytic enzymes remains unclear. Here, we report that the WalR-WalK (WalRK) two-component signaling pathway functions in the homeostatic control of the cell wall hydrolases required for expansion of the PG during growth.

The WalRK two-component system (TCS) was discovered over two decades ago and is among the most broadly conserved TCS in Firmicutes (*Fabret and Hoch, 1998*; *Dubrac et al., 2008a*; *Takada et al., 2018*). WalK (also referred to as YycG, VicK, or MicA) is a membrane-anchored sensor kinase and WalR (also called YycF, VicR, or MicB) is a DNA binding response regulator of the OmpR family (*Fabret and Hoch, 1998*; *Dubrac et al., 2008a*; *Dubrac and Msadek, 2008b*; *Okajima et al., 2008*). In most bacteria that encode this TCS, two additional genes, *walH* (*yycH*) and *walI* (*yycI*), reside in an operon with them. In *B. subtilis*, WalH and WalI are negative regulators of the WalK sensor kinase and the three integral membrane proteins assemble into a multimeric complex (*Szurmant et al., 2005*; *Szurmant et al., 2007*; *Szurmant et al., 2008*). The WalRK system is essential in most Firmicutes, making it an attractive antibiotic target (*Barrett and Hoch, 1998*; *Gotoh et al., 2010*). Accordingly, the WalRK pathway has been extensively studied in *B. subtilis* as well as several important Gram-positive pathogens. In all cases where it has been examined, the WalR regulon contains genes encoding cell wall hydrolases (*Bisicchia et al., 2007*; *Howell et al., 2003*; *Ahn and Burne, 2007*; *Ng et al., 2005*; *Liu, 2006*; *Dubrac et al., 2007*). Furthermore, cells engineered to constitutively express a subset of these enzymes can bypass the essentiality of the signaling pathway (*Ng et al., 2003*; *Delaune et al., 2011*; *Takada et al., 2018*). These findings have

led to the view that the essential role of WalRK is to coordinate cell wall metabolism with growth. However, despite two decades of research, what the WalK sensor kinases senses and how this pathway functions in cell wall homeostasis have remained mysterious.

In *B. subtilis*, phosphorylated WalR (WalR ~P) controls the synthesis of several cell wall hydrolases; among them are two enzymes (CwlO and LytE) that are critical for cell wall elongation. Cells lacking either PG hydrolase are viable but depletion of one in the absence of the other causes a cessation of growth followed by lysis (*Bisicchia et al., 2007*; *Hashimoto et al., 2012*). Both enzymes are D,L-endopeptidases and cleave the peptide bond between the second (γ-D-Glu) and third (mDAP) amino acid in the stem peptide of PG (*Ishikawa et al., 1998*; *Yamaguchi et al., 2004*). CwlO is controlled by a membrane complex composed of the non-canonical ABC transporter FtsEX and two integral membrane proteins SweC and SweD (*Meisner et al., 2013*; *Domínguez-Cuevas et al., 2012*; *Brunet et al., 2019*). LytE is a secreted enzyme with LysM domains that direct it to the lateral cell wall (*Margot et al., 1998*; *Ishikawa et al., 1998*; *Buist et al., 2008*). How the level and activity of these essential elongation hydrolases are regulated remains incompletely understood. Here, we report that the WalK sensor kinase monitors the activity of CwlO and LytE by sensing the cleavage products generated by them. In response, WalK controls WalR-dependent changes in the expression and activity of these enzymes. Thus, this essential two component system functions in the homeostatic control of PG hydrolysis required for growth. This represents the first homeostatic pathway for cell wall hydrolysis in bacteria and we propose that cell wall remodeling in related Gram-positive pathogens is subject to similar regulatory control.

## Results

### LytE levels increase in the absence of CwlO maintaining cell envelope integrity

In the course of characterizing LytE protein levels in various mutant backgrounds, we discovered that LytE levels increase approximately 2-fold in the absence of CwlO (*Figure 1A*). To determine whether this increase was due to changes in *lytE* transcription, we fused the *lytE* promoter to *lacZ* and compared ß-galactosidase activity in wild-type and cells lacking CwlO. As can be seen in *Figure 1B*, transcription from the $P_{lytE}$ promoter increased ~2 fold in the Δ*cwlO* mutant. A similar increase in $P_{lytE}$ transcription was observed in cells lacking the FtsEX complex, which is required for CwlO activity (*Figure 1B*) (*Meisner et al., 2013*). Furthermore, a point mutation in the Walker A motif in FtsE, predicted to impair ATP binding (*Yang et al., 2011*; *Meisner et al., 2013*) but not CwlO association with FtsX (*Brunet et al., 2019*) also resulted in increased *lytE* transcription (*Figure 1—figure supplement 1*). From these experiments we conclude that cells lacking CwlO activity increase expression of *lytE*. We also observed a modest but reproducible increase in *lytE* transcription in cells lacking LytE (*Figure 1B*), suggesting that *B. subtilis* increases *lytE* expression in response to reduction in D,L-endopeptidase activity in general.

To investigate whether the ~2 fold change in LytE levels in the Δ*cwlO* mutant has any physiological consequences, we used a strain lacking both *cwlO* and *lytE* that contained an IPTG-regulated allele of *lytE*. First, we determined the inducer concentration that resulted in LytE levels equivalent to wild-type (*Figure 1A*) and then examined the cells by fluorescence microscopy (*Figure 1C*, *Figure 1—figure supplement 2*). As reported previously, cells lacking *cwlO* were shorter and fatter than wild-type and these morphological phenotypes were largely homogenous throughout the population (*Hashimoto et al., 2012*; *Meisner et al., 2013*; *Brunet et al., 2019*). Furthermore, based on cytoplasmic mCherry fluorescence (*Figure 1C*) and propidium iodide staining (*Figure 1D*), the Δ*cwlO* mutant cells had intact membranes. By contrast, cells lacking *cwlO* in which LytE was artificially maintained at wild-type levels had heterogeneous morphologies with >20% lysis or membrane permeability defects (*Figure 1C and D*, *Figure 1—figure supplement 2*). These results indicate that the increase in LytE levels allows the Δ*cwlO* mutant to maintain membrane integrity and a homogenous morphology. Thus, these data suggest that cells lacking CwlO compensate for the reduction in D,L-endopeptidase activity by increasing expression of a second D,L-endopeptidase, LytE.

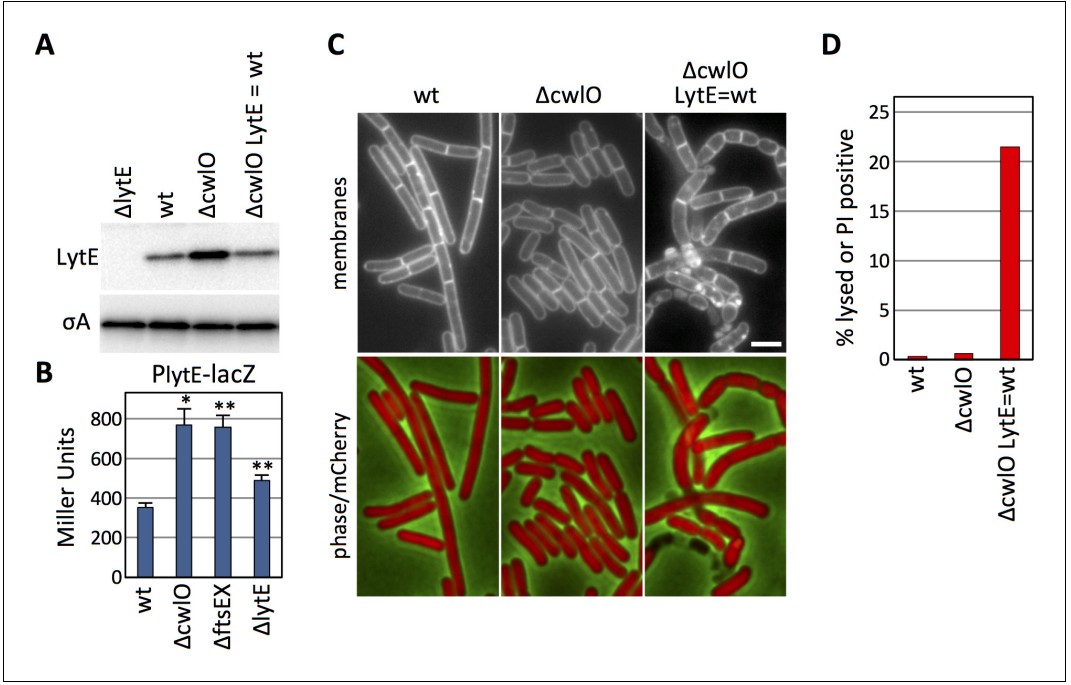

**Figure 1.** B. *subtilis* increases *lytE* expression in the absence of CwlO activity to maintain cell envelope integrity. (**A**) Immunoblot analysis of LytE produced under the control of its native promoter or under IPTG control. The indicated strains (Δ*lytE*, wild-type (wt), Δ*cwlO*, and Δ*cwlO* Δ*lytE* P(IPTG)-*lytE* (LytE = wt)) were grown in CH medium with or without 500 µM IPTG and harvested at an OD600 ~0.4. SigA protein levels were analyzed to control for loading. (**B**) Bar graph showing β-galactosidase activity from a *lytE* promoter (P*lytE*) fusion to *lacZ* in wild-type (wt), Δ*cwlO*, Δ*ftsEX*, and Δ*lytE* strains. Activity was assayed in exponentially growing cultures in LB. Error bars represent standard deviation from three biological replicates. Asterisks indicate p-values calculated using Welch's unequal variances *t*-test compared to wildtype (*<0.02, **<0.005). (**C**) Representative fluorescent images of strains from (**A**) harboring cytoplasmic mCherry grown under identical conditions as in (**A**). Membranes were visualized with TMA-DPH (top), and merged images of cytoplasmic mCherry and phase-contrast are shown (bottom). Scale bar indicates 2 µm. (**D**) Bar graph showing the percentage of cells with envelope integrity defects in wild-type (wt), Δ*cwlO*, and a Δ*cwlO* mutant in which LytE levels are held at levels equivalent to wild-type (LytE = wt). Cells without cytoplasmic fluorescence and/or that stained with propidium iodide were scored as lysed or PI positive. >500 cells were scored per strain. The images and immunoblots in this figure were representatives from three independent experiments.

The online version of this article includes the following source data and figure supplement(s) for figure 1:

**Source data 1.** *Figure 1* B-galactosidase assay Miller Units.
**Figure supplement 1.** Cells harboring an *ftsE* Walker A mutation increases *lytE* transcription.
**Figure supplement 1—source data 1.** *Figure 1—figure supplement 1* B-galactosidase assay Miller Units.
**Figure supplement 2.** *B. subtilis* increases LytE levels to maintain cell envelope integrity.

## B. subtilis modulates *lytE* transcription in response to changes in D,L-endopeptidase activity

The experiments described above indicate that B. *subtilis* increases *lytE* expression in the absence of CwlO activity. We next investigated whether cells decrease *lytE* transcription in the presence of too much D,L-endopeptidase activity. Because CwlO is regulated post-translationally by FtsEX (*Domínguez-Cuevas et al., 2012*; *Meisner et al., 2013*), we used LytE to generate high D,L-endopeptidase activity. We introduced a strong IPTG-inducible promoter fusion to *lytE* and a catalytic mutant (C247S) into a strain harboring our P*lytE*-*lacZ* reporter and monitored ß-galactosidase activity after induction. Strikingly, cells with increased levels of wild-type LytE had reduced *lytE* transcription, while cells over-expressing the catalytic mutant had P*lytE* promoter activity similar to wild-type (*Figure 2A and B*).

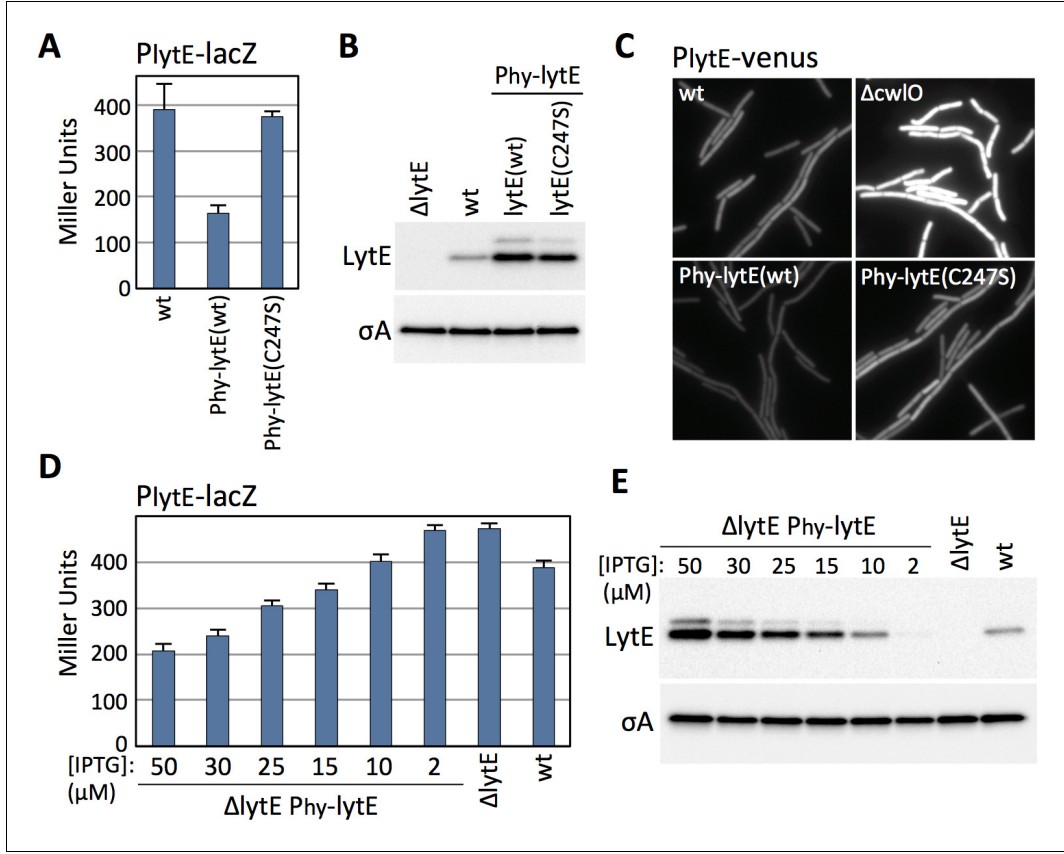

**Figure 2.** *B. subtilis* modulates *lytE* transcription in response to changes in D,L-endopeptidase activity. (**A**) Bar graph showing β-galactosidase activity from the P$_{lytE}$-*lacZ* reporter in wild-type (wt), and strains harboring the *lytE* gene or a catalytic mutant (C247S) under the control of the strong IPTG-inducible promoter P$_{hyperspank}$ (P$_{hy}$) and with an optimized ribosome binding site. Activity was assayed in exponentially growing cultures 60 min after the addition of 50 µM IPTG. Error bars represent standard deviation from three biological replicates. (**B**) Immunoblot analysis of LytE in the same strains as in (**A**), 60 min after induction with 50 µM IPTG. SigA protein levels were analyzed to control for loading. (**C**) Representative images of cytoplasmic Venus fluorescence from the P$_{lytE}$-*venus* reporter in the indicated strains visualized 30 min after addition of 50 µM IPTG. (**D**) Bar graph showing β-galactosidase activity from P$_{lytE}$-*lacZ* in the indicated strains. Cells were grown in LB medium or LB medium supplemented with the indicated concentrations of IPTG. β-galactosidase activity was assayed 60 min after induction of *lytE*. Error bars represent standard deviation from three biological replicates. (**E**) Immunoblot analysis of LytE and SigA in the strains used in (**D**). All representative images and immunoblots in this figure are from one of three independent experiments.

The online version of this article includes the following source data and figure supplement(s) for figure 2:

**Source data 1.** *Figure 2A* B-galactosidase assay Miller Units.
**Source data 2.** *Figure 2D* B-galactosidase assay Miller Units.
**Figure supplement 1.** Expression from the IPTG-regulated promoter (P$_{hyperspank}$) is homogenous across all cells in the population.

Immunoblot analysis and *lacZ* reporters are population-based assays. To address whether the changes in *lytE* transcription in response to high or low D,L-endopeptidase activity were homogenous throughout the population, we built a *lytE* promoter fusion to the gene encoding the yellow fluorescent protein variant Venus (P$_{lytE}$-*venus*). The fluorescent reporter was introduced into strains with high or low D,L-endopeptidase activity and then monitored during exponential growth by fluorescence microscopy. As can be seen in *Figure 2C*, all cells lacking *cwlO* had increased Venus fluorescence while all cells over-expressing *lytE* had reduced fluorescence. Collectively, these results indicate that *B. subtilis* modulates *lytE* expression in response to both an increase and decrease in D,L-endopeptidase activity.

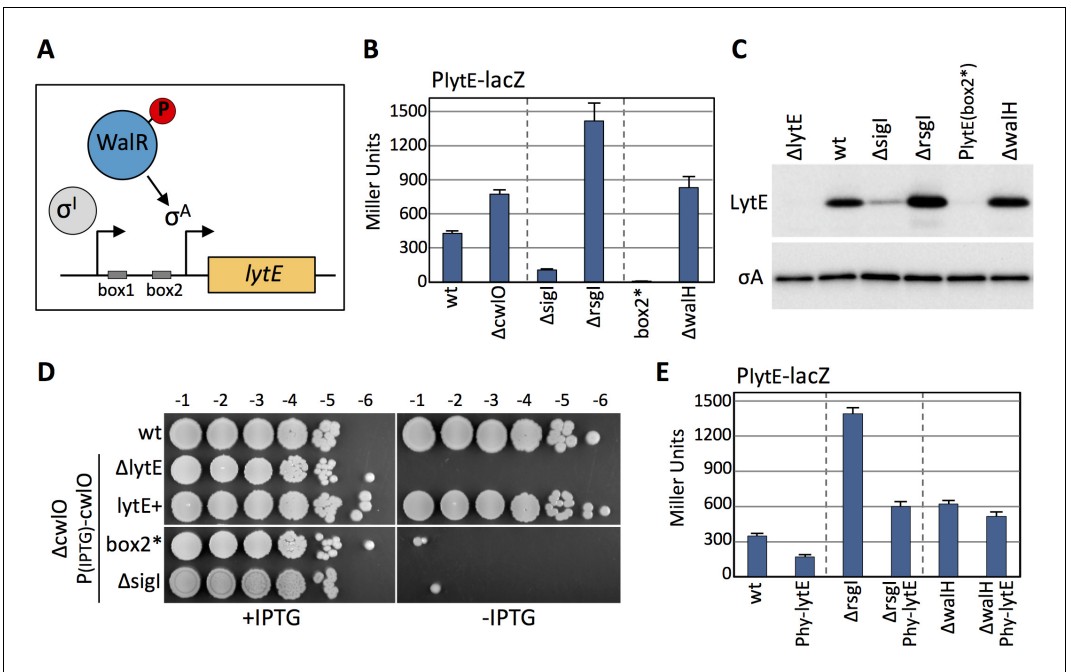

**Figure 3.** Modulation of lytE transcription in response to changes in D,L-endopeptidase activity requires WalR. (**A**) Schematic diagram of the *lytE* gene with its two promoters recognized by SigI and SigA. Phosphorylated WalR binds to two sites (box 1 and 2) flanking the SigA promoter. (**B**) Bar graph showing β-galactosidase activity from the $P_{lytE}$-*lacZ* reporter in the indicated strains. Activity was assayed in exponentially growing cultures in LB medium. Error bars represent standard deviation from three biological replicates. (**C**) Immunoblot analysis of LytE protein levels in the strains used in (**B**). SigA levels were analyzed to control for loading. Immunoblots from one of two biological replicates are shown. (**D**) Spot dilutions of the indicated strains in the presence and absence of *cwlO* expression. Strains were grown in the presence of IPTG (500 μM) to $OD_{600}$ ~2.0. The cultures were washed twice without inducer, normalized to an $OD_{600}$ = 1.5, 10-fold serially diluted, and spotted (5 μL) onto LB agar plates with or without 500 μM IPTG. Representative plates from one of two biological replicates are shown. (**E**) Bar graph showing β-galactosidase activity from the $P_{lytE}$-*lacZ* reporter in indicated strains. Strains harboring the strong IPTG-inducible promoter fusion to *lytE* ($P_{hy}$-*lytE*) were induced for 60 min with 50 μM IPTG. Error bars represent standard deviation from three biological replicates.

The online version of this article includes the following source data for figure 3:

**Source data 1.** *Figure 3B* B-galactosidase assay Miller Units.
**Source data 2.** *Figure 3E* B-galactosidase assay Miller Units.

The experiments presented thus far indicate that *B. subtilis* can increase or decrease *lytE* transcription in cells lacking *cwlO* or over-expressing *lytE*. However, it seemed unlikely that *B. subtilis* evolved a mechanism to compensate for gene deletion and over-expression. To investigate whether *B. subtilis* modulates *lytE* transcription in response to more physiological changes in D,L-endopeptidase activity, we used an IPTG-regulated allele of *lytE* and grew cells at different inducer concentrations to produce a range of LytE levels that were both above and below wild-type levels (*Figure 2E*). As can been seen in *Figure 2D*, we found that $P_{lytE}$ transcription inversely correlated with the amount of D,L-endopeptidase produced. Importantly, using the same IPTG-regulated promoter fused to *gfp*, we found that at IPTG concentrations similar to those used to express *lytE* all cells in the population had equivalent GFP fluorescence (*Figure 2—figure supplement 1*). Thus, the graded response to the changes in LytE levels observed in the ensemble assays reflect similar changes in D, L-endopeptidase levels in all cells in the population.

## Modulation of *lytE* expression requires WalR and not SigI

The *lytE* gene is has two promoters that influence each other (*Tseng et al., 2011*; *Salzberg et al., 2013*) (*Figure 3A*). One is recognized by the alternative sigma factor Sigma I (SigI) and the other is

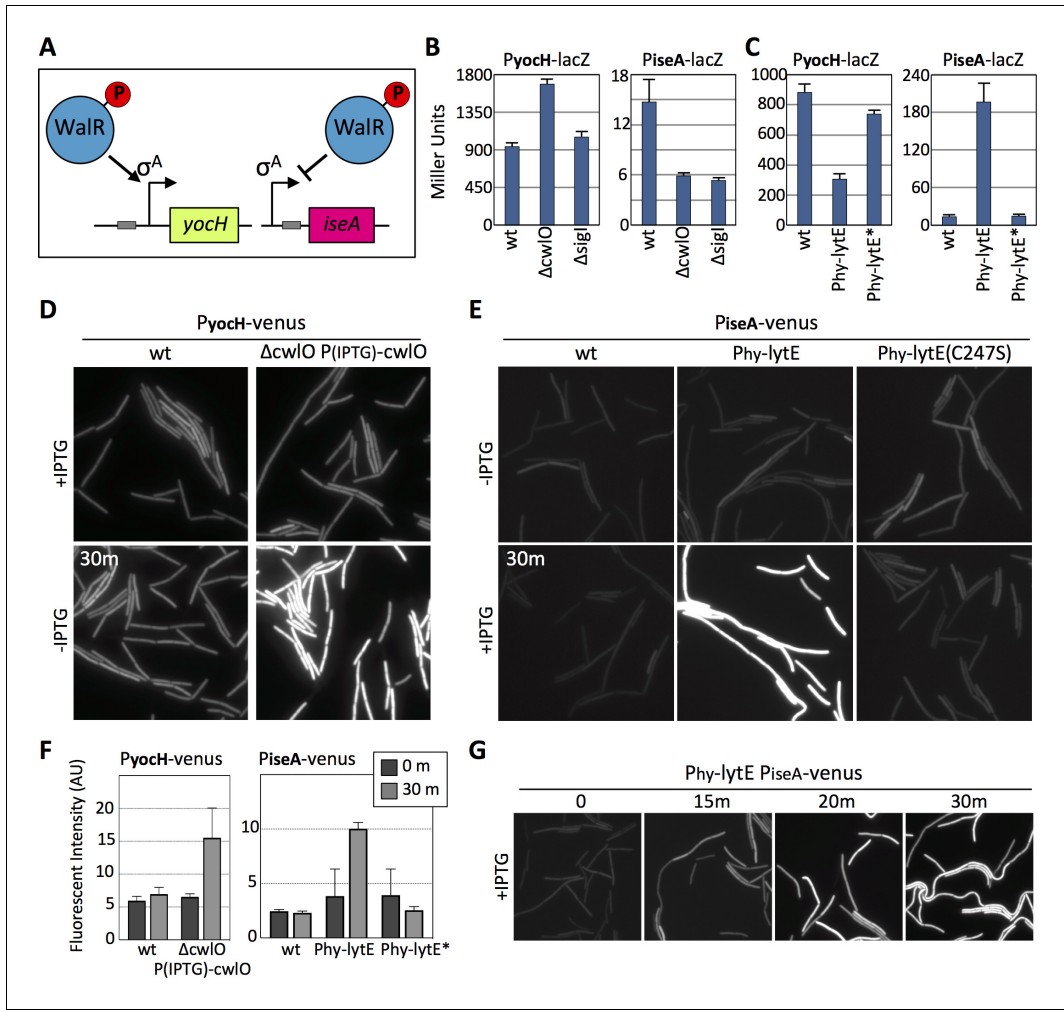

**Figure 4.** The WalRK signaling pathway responds to changes in D,L-endopeptidase activity. (**A**) Schematic diagrams of the *yocH* and *iseA* genes. Phosphorylated WalR activates transcription of *yocH* and represses transcription of *iseA*. The positions of the WalR binding sites (gray boxes) are shown. (**B and C**) Bar graph showing β-galactosidase activity from P*yocH*-*lacZ* and P*iseA*-*lacZ* reporters in the indicated strains. Activity was assayed in exponentially growing cultures in LB medium. Strains harboring strong IPTG-inducible promoter fusion to *lytE* (P$_{hy}$-*lytE*) and the LytE catalytic mutant (C247S) (P$_{hy}$-*lytE**) were induced for 60 min with 50 µM IPTG. Error bars represent standard deviation from three biological replicates. (**D**) Representative fluorescent images of the indicated strains harboring P*yocH*-*venus*. Cells were grown to OD$_{600}$ ~0.3 in LB medium supplemented with 1 mM IPTG, washed in medium lacking inducer and imaged before and 30 min after growth in LB medium lacking inducer. (**E**) Representative fluorescent images of the indicated strains harboring P*iseA*-*venus*. Strains were grown to OD$_{600}$ ~0.3 in LB medium and imaged before and 30 min after addition of IPTG (50 µM). Representative images are from one of three independent experiments. (**F**) Quantification of the average fluorescence intensity, normalized to cell area, of strains shown in (**D**) and (**E**).>1000 cells were analyzed from three independent experiments. Dark gray bars (0 m) are from cells prior to IPTG removal for the CwlO depletion experiment monitoring P*yocH*-*venus* and prior to IPTG addition for the *lytE* over-expression experiment monitoring P*iseA*-*venus*. Light gray bars (30 m) are from cells 30 min after IPTG removal (left graph) and 30 after IPTG addition (right graph). (**G**) Representative fluorescent images of P*iseA*-*venus* at the indicated times after addition of IPTG (50 µM). P*iseA*-*venus* is de-repressed within 15 min after induction of *lytE*.

The online version of this article includes the following source data and figure supplement(s) for figure 4:

**Source data 1.** *Figure 4B* B-galactosidase assay Miller Units.
**Source data 2.** *Figure 4C* B-galactosidase assay Miller Units.
**Source data 3.** *Figure 4F* cwlO depletion, cellular fluorescence intensity.
**Source data 4.** *Figure 4F* lytE over-expression, cellular fluorescence intensity.
**Figure supplement 1.** Validation of P$_{iseA}$-*venus* and P$_{yocH}$-*venus* reporters.

*Figure 4 continued on next page*

*Figure 4 continued*

**Figure supplement 2.** P$_{yocH}$-*venus* transcription decreases in response to LytE over-expression.

**Figure supplement 3.** The P$_{iseA}$-*venus* and P$_{yocH}$-*venus* reporters are not influenced by SigI.

**Figure supplement 4.** CwlO levels are modulated in response to changes in D,L-endopeptidase activity.

**Figure supplement 5.** PrkC is not involved in the response to changes in D,L-endopeptidase activity.

---

controlled by Sigma A (SigA) but requires the phosphorylated form of the response regulator WalR (*Salzberg et al., 2013*). Cells lacking SigI have reduced P$_{lytE}$-*lacZ* expression and reduced LytE protein levels, while cells lacking the anti-SigI factor RsgI have increased *lytE* transcription and increased LytE protein levels (*Figure 3B and C*). Similarly, a point mutation in one of the two WalR binding sites in the *lytE* promoter (box2\*) (*Salzberg et al., 2013*) abolishes *lytE* transcription and LytE protein levels (*Figure 3B and C*). Furthermore, cells lacking WalH, a negative regulator of the WalRK two-component system (*Szurmant et al., 2005*), results in increased *lytE* transcription and LytE protein levels (*Figure 3B and C*). Consistent with these observations and the synthetic lethal relationship between *lytE* and *cwlO*, cells lacking SigI or harboring a point mutation in the WalR binding site in the *lytE* promoter are inviable when CwlO is depleted (*Salzberg et al., 2013*) (*Figure 3D*). To investigate whether the RsgI-SigI or the WalRK signaling pathway is involved in the response to changes in D,L-endopeptidase activity, we monitored the P$_{lytE}$-*lacZ* response to increased LytE levels in Δ*rsgI* and Δ*walH* mutants in which SigI and WalR were constitutively active. As can be seen in *Figure 3E*, over-expression of *lytE* in the Δ*rsgI* mutant resulted in reduced *lytE* transcription, while in the Δ*walH* mutant *lytE* transcription was unchanged. These data suggest that the WalRK signaling pathway is responsible for mediating the observed response to changes in D,L-endopeptidase activity.

## The WalRK signaling pathway responds to changes in D,L-endopeptidase activity

To further test whether the WalRK pathway responds to changes in D,L-endopeptidase activity, we generated transcriptional reporters for two well-characterized genes (*yocH* and *iseA*) that are specifically regulated by WalR and not SigI (*Bisicchia et al., 2007*). *yocH* is positively regulated by phosphorylated WalR (WalR ~P) while *iseA* is negatively regulated by WalR ~P (*Figure 4A*). We fused both promoters to *lacZ* and separately to *venus*. To validate these reporters, we monitored their activity in strains lacking WalH or WalI, two negative regulators of WalK (*Szurmant et al., 2005*; *Szurmant et al., 2007*). In the absence of either, WalR activity is high and P$_{yocH}$-*venus* transcription increased, while P$_{iseA}$-*venus* transcription decreased (*Figure 4—figure supplement 1A and B*). Furthermore, and as expected, depletion of WalRK resulted in strong de-repression of P$_{iseA}$-*venus* (*Figure 4—figure supplement 1C*).

Next, we used these WalR-specific reporters to investigate whether the WalRK pathway responds to changes in D,L-endopeptidase activity. When monitored by ß-galactosidase assay, P$_{yocH}$ transcription increased in the absence of CwlO and decreased when *lytE*, but not *lytE*(C247S), was over-expressed (*Figure 4B and C*). Reciprocally, P$_{iseA}$ transcription was reduced in the absence of CwlO and was strongly de-repressed when *lytE* was over-expressed (*Figure 4B and C*). Similar results were obtained with our fluorescent reporters. Within 30 min after shutting off *cwlO* transcription, P$_{yocH}$-directed transcription of *venus* increased (*Figure 4D*). Furthermore, 15 min after inducing *lytE* transcription, we could detect de-repression of the P$_{iseA}$-*venus* reporter (*Figure 4E and G*), and after 30 min down-regulation of the P$_{yocH}$-*venus* was apparent (*Figure 4—figure supplement 2*). Quantification of Venus fluorescence (*Figure 4F*) indicates that our fluorescent and *lacZ* reporters respond similarly to changes in D,L-endopeptidase levels, although for unknown reasons the magnitude of P$_{iseA}$ de-repression was not as great with the fluorescent reporter. Importantly, these WalR-specific promoters were not affected by a deletion of *sigI* (*Figure 4B and C* and *Figure 4—figure supplement 3A and B*), nor were their dynamics altered in the Δ*sigI* mutant in response to changes in D,L-endopeptidase activity (*Figure 4—figure supplement 3C*).

It is noteworthy that *both lytE* and *cwlO* are expressed under the control of WalR ~P. Thus, changes in D,L-endopeptidase activity should not only impact the levels of LytE, as shown in *Figures 1* and *2*, but also the levels of CwlO. Immunoblots to monitor the levels of CwlO in cells over-expressing *lytE* indicate that this is indeed the case (*Figure 4—figure supplement 4A*).

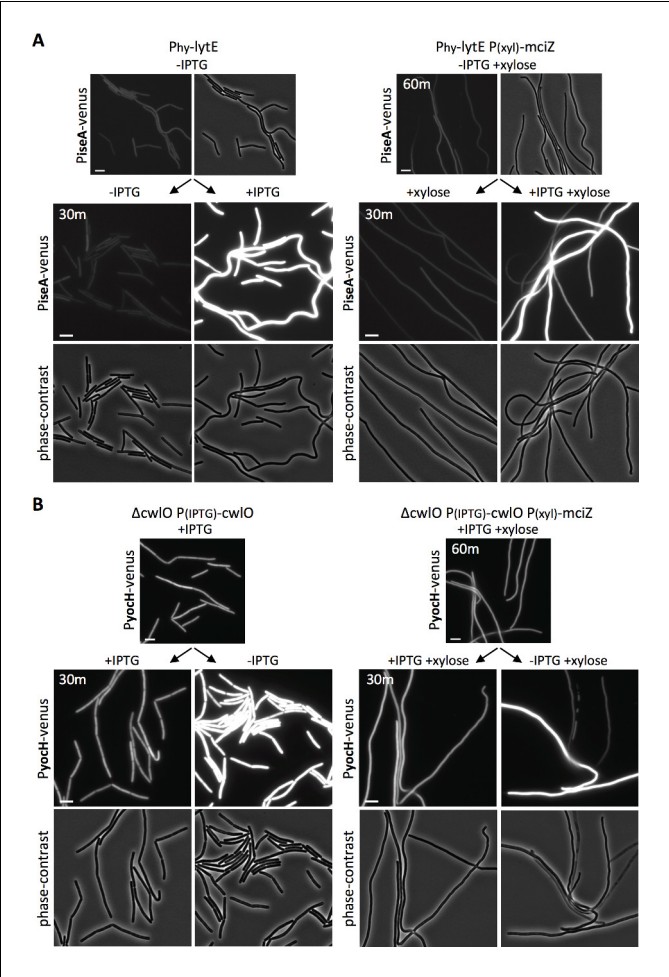

**Figure 5.** WalRK responds to changes in D,L endopeptidase activity in cells inhibited for divisome assembly. (**A**) Representative fluorescence and phase-contrast images of cells harboring P$_{iseA}$-*venus* before and 30 min after induction of *lytE* with 50 µM IPTG. Images on the right are from a strain that also contained the FtsZ inhibitor MciZ under xylose control (P$_{(xyl)}$-*mciZ*) and were grown in the presence of 10 mM xylose for 60 min prior to IPTG addition. (**B**) Representative fluorescence and phase-contrast images of cells harboring P$_{yocH}$-*venus* before and 30 min after removal of IPTG to shut off *cwlO* transcription. Images on the right are from a strain that also contained P$_{(xyl)}$-*mciZ* and was grown in the presence of 10 mM xylose for 40 min, prior to IPTG removal. A subset of the division-inhibited cells did not de-repress P$_{iseA}$-*venus* (**A**) or induce P$_{yocH}$-*venus* (**B**); this could be due to loss of viability (see *Figure 5—figure supplements 2* and *3*). Representative images are from one of three independent experiments. Scale bar indicates 5 µm.

The online version of this article includes the following figure supplement(s) for figure 5:

**Figure supplement 1.** MciZ expression disrupts divisome assembly.
**Figure supplement 2.** Loss of membrane integrity in cells inhibited for divisome assembly.
**Figure supplement 3.** Loss of membrane integrity in cells inhibited for divisome assembly.

---

Furthermore, cells lacking *lytE* have a modest but reproducible increase in CwlO (*Figure 4—figure supplement 4B*). Taken together with the data in *Figure 3E*, these results suggest that the WalRK signaling pathway monitors the extent of D,L endopeptidase activity and modulates *lytE* and *cwlO* transcription in response.

Recent studies indicate that the serine/threonine kinase PrkC also controls WalR activity (*Libby et al., 2015*). PrkC principally regulates WalR in stationary phase and therefore was unlikely to mediate the response to changes in D,L-endopeptidase activity during exponential growth observed here. However, to directly test this we analyzed P$_{iseA}$ and P$_{yocH}$ transcription in wild-type and the ΔprkC mutant before and after inducing *lytE* (*Figure 4—figure supplement 5*). As

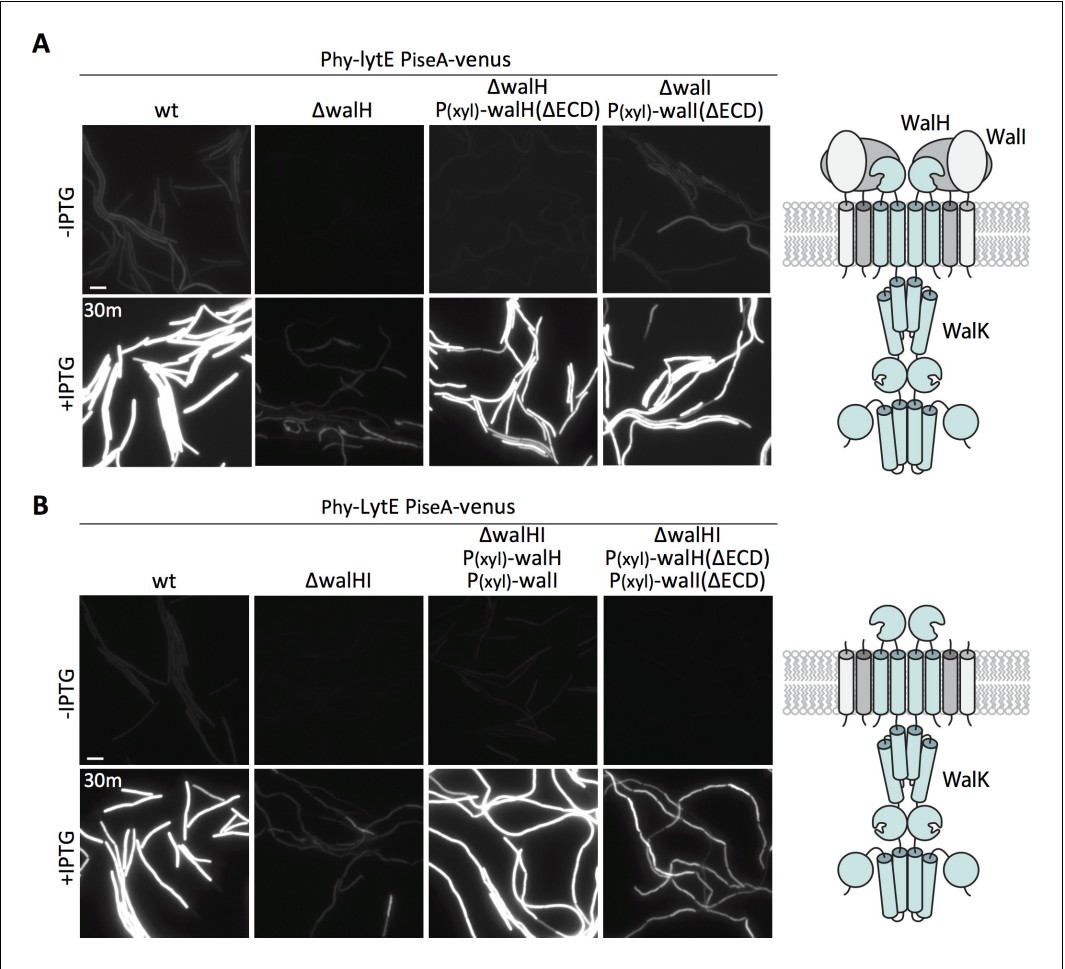

**Figure 6.** The extracellular domains of WalH and WalI are not required for WalK to respond to changes in D,L-endopeptidase activity. (**A**) Representative fluorescence images of the indicated strains harboring $P_{iseA}$-*venus* and a strong IPTG-inducible promoter fused to *lytE* ($P_{hy}$-*lytE*). Strains were grown to $OD_{600}$ ~0.3 in LB medium and imaged before and 30 min after addition of 50 µM IPTG. The medium was supplemented with 10 mM xylose for the strains harboring xylose-regulated alleles of *walH* or *walI* with deletions of their extracellular domains (ΔECD). Schematic diagram of the putative WalK/WalH/WalI membrane complex is shown to the right. (**B**) Representative images of $P_{iseA}$-*venus* expression before and 30 min after *lytE* over-expression in strain lacking both *walH* and *walI* and complemented by xylose-induced full-length genes or ΔECD deletion variants. Schematic diagram of the putative membrane complex with WalH and WalI lacking their ECDs is shown on the right. Representative images are from one of three independent experiments. Scale bar indicates 5 µm.

The online version of this article includes the following figure supplement(s) for figure 6:

**Figure supplement 1.** Complementation of Δ*walH* and Δ*walI* with xylose-regulated alleles.

anticipated, the transcriptional responses were similar in the presence and absence of PrkC, ruling out a role for this signaling kinase in responding to D,L-endopeptidase activity.

## The WalK sensor kinase responds to changes in D,L-endopeptidase activity in the absence of cell division

Previous work on the WalRK pathway indicates that WalK, but not WalH or WalI, localizes to the divisome and does so in an FtsZ-dependent manner (*Fukushima et al., 2008*). Furthermore, depletion of FtsZ or other divisome components resulted in a decrease in *yocH* mRNA levels and an increase in *pdaC* transcript levels, consistent with a reduction in WalR activity (*Fukushima et al., 2008*; *Fukushima et al., 2011*). These studies formed the basis of the prevailing model for WalRK function in which WalK signaling and WalR-dependent expression of cell wall hydrolases are linked to growth

via the divisome (*Fukushima et al., 2008*; *Dubrac et al., 2008a*; *Fukushima et al., 2011*). Specifically, it was proposed that in non-dividing cells WalK is held inactive along the lateral membranes by its negative regulators WalH and WalI, while in actively growing and dividing cells, WalK localizes to the septum without its inhibitors where it is competent to activate WalR. Our findings that WalK responds to changes in PG hydrolases that act along the lateral cell wall appear inconsistent with a model in which WalK is only active at the septum. To address this discrepancy, we investigated whether the WalRK pathway could respond to changes in D,L-endopeptidase activity in cells lacking a divisome.

To prevent divisome assembly, we took advantage of the FtsZ inhibitor MciZ (*Handler et al., 2008*). Cells harboring a xylose-regulated allele of *mciZ* were grown for 60 min in the presence of inducer to block FtsZ-ring assembly (*Figure 5—figure supplement 1*). The resulting filaments were then induced to express *lytE*, and WalRK activity was monitored 30 min later using our P$_{iseA}$-venus reporter. As can be seen in *Figure 5A*, *lytE* over-expression resulted in similar de-repression of P$_{iseA}$-venus in wild-type and cells lacking a divisome. Reciprocally, depletion of CwlO in filamenting cells for 30 min triggered induction of P$_{yocH}$-venus to the same extent as CwlO depletion in wild-type cells (*Figure 5B*). We note that a subset of the filaments in both experiments did not respond to changes in D,L-endopeptidase activity. We suspect this is due to a loss in viability, which is largely consistent with propidium iodide staining (*Figure 5—figure supplements 2* and *3*). Altogether, our data indicate that the WalRK signaling pathway is capable of responding to changes in PG hydrolase activity in the absence of division, and therefore argue that WalK can function along the lateral membranes.

## WalK responds to changes in D,L-endopeptidase activity in the absence of the extracellular domains of the WalH and WalI regulators

The two WalK inhibitors, WalH and WalI, are single-pass integral membrane proteins with large extracellular domains (ECDs) (*Szurmant et al., 2005*) (*Figure 6A* - schematic). Both regulators reside in a membrane complex with WalK (*Szurmant et al., 2007*) and could therefore function in signal recognition. Previous work from Szurmant and Hoch showed that basal WalRK activity was unaffected by deletions of the ECDs of WalH and WalI (*Szurmant et al., 2008*). To investigate whether either of the extracellular domains were required for WalRK signaling in response to changes in D,L-endopeptidase activity, we sought to test strains in which these domains were deleted. We generated xylose-regulated alleles of full-length *walH* and *walI* and deletion variants (*walH*(Δ61–455) and *walI*(Δ36–280)) that were identical to those used previously (*Szurmant et al., 2008*). We introduced these alleles at an ectopic genomic locus in strains lacking *walH* or *walI* that harbored the P$_{iseA}$-venus reporter. As expected, in the absence of xylose, the Δ*walH* and Δ*walI* strains had virtually undetectable P$_{iseA}$-venus expression, indicative of high WalRK activity and strong repression of the P$_{iseA}$ promoter (*Figure 6A*, *Figure 6—figure supplement 1*). Similar to what was reported previously, expression of both the full-length and truncated alleles of *walH* and *walI* restored basal P$_{iseA}$-venus transcription (*Figure 6A*, *Figure 6—figure supplement 1*). Importantly, cells harboring the full-length and truncated alleles of *walH* and *walI* responded similarly to wild-type when *lytE* was over-expressed (*Figure 6A*, *Figure 6—figure supplement 1*). In all cases, P$_{iseA}$-venus transcription was de-repressed, indicative of a reduction in WalRK signaling. These data indicate that the extracellular domains of WalH and WalI are not individually required for WalRK signaling in response to changes in D,L-endopeptidase activity. To test whether cells lacking the extracellular domains of *both* WalH and WalI can respond to *lytE* over-expression, we built stains harboring xylose-regulated alleles of both *walH* and *walI* or their deletion variants. As can be seen in *Figure 6B*, in both strains P$_{iseA}$-venus transcription was de-repressed upon over-expression of *lytE*, although the response was more modest in the strain harboring the deletion variants. Collectively, these data argue that the extracellular domains of WalH and WalI are not necessary for WalRK signaling in response to changes in D,L-endopeptidase activity and suggest that the extracellular domain of WalK functions as the sensor.

## The WalK extracellular domain is required to respond to changes in D,L-endopeptidase activity

WalK contains a 148 amino acid extracellular loop that is homologous to Per-Arnt-Sim (PAS)-like domains, also known as an sCache domain (*Upadhyay et al., 2016*) (*Figure 6A* – schematic). These domains are a common feature of sensor kinases and in some cases have been shown to bind

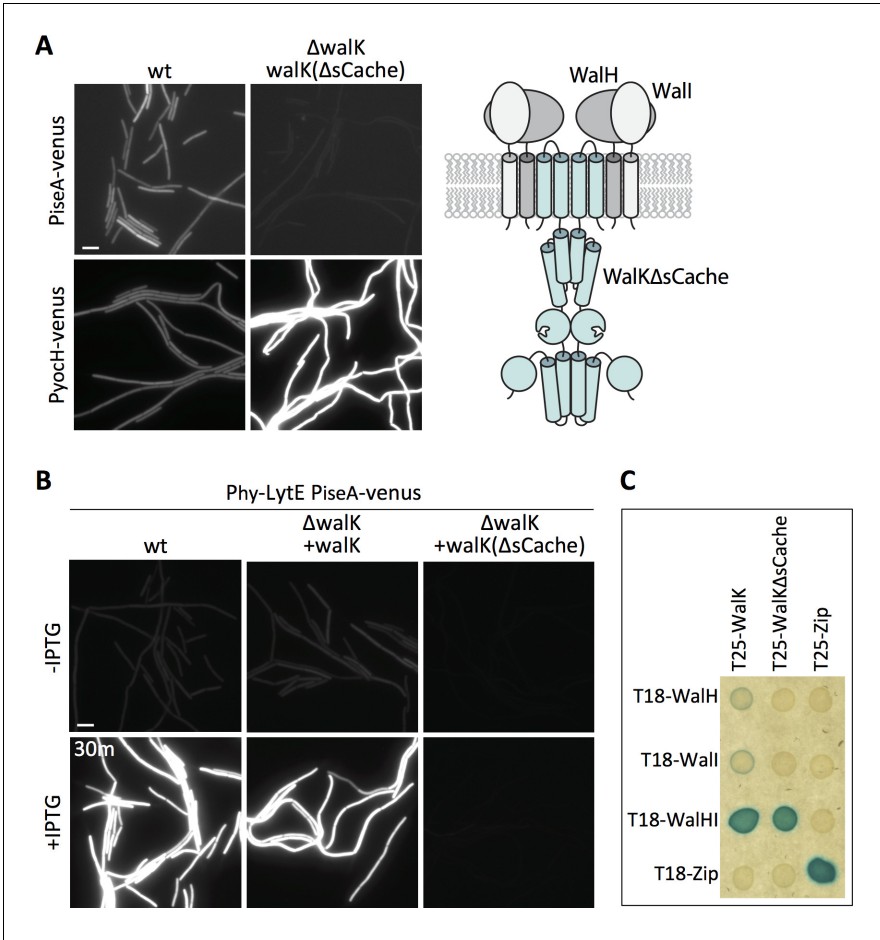

**Figure 7.** Deletion of the extracellular sCache domain of WalK renders the sensor kinase unresponsive to changes in D,L-endopeptidase activity. (A) Representative fluorescence images of the indicated strains harboring $P_{iseA}$-venus or $P_{yocH}$-venus reporters. Cells were grown to $OD_{600}$ ~0.4 in LB medium. Strains in which WalK lacks its extracellular sensory domain (ΔsCache) have high WalRK activity, leading to strong repression of $P_{iseA}$-venus and elevated transcription of $P_{yocH}$-venus. Schematic model of the signaling complex with WalK lacking its extracellular domain is shown on the right. (B) Representative fluorescence images of the indicated strains harboring $P_{iseA}$-venus and $P_{hy}$-lytE. Strains were grown to $OD_{600}$ ~0.3 in LB medium and imaged before and 30 min after addition of 50 µM IPTG. Representative images are from one of three independent experiments. Scale bar indicates 5 µm. (C) WalK and WalKΔsCache interact with WalH and WalI in the bacterial adenylate cyclase two-hybrid (BACTH) assay. The BTH101 *E. coli* reporter strain containing plasmids expressing the indicated protein fusions to the complementing (T18 and T25) domains of the *Bordetella* adenylate cyclase. The T18-WalHI plasmid contains a T18-WalH fusion and untagged WalI. Cells were grown to early stationary phase in LB at 37°C, normalized to $OD_{600}$ = 0.2, and 3 µL of each was spotted on LB agar plates supplemented with X-Gal and IPTG. Plates were incubated overnight at 30°C. The 'Zip' fusions, composed of the leucine zipper domain of GCN4, served as positive and negative controls.

The online version of this article includes the following figure supplement(s) for figure 7:

**Figure supplement 1.** Deletion of the extracellular sCache domain of WalK renders the sensor kinase unresponsive to depletion of CwlO.

signaling ligands (*Chang et al., 2010*; *Henry and Crosson, 2011*; *Upadhyay et al., 2016*). Previous studies indicate that deletion of the extracellular sCache domain of WalK (WalKΔsCache) results in constitutive signaling and high WalR activity (*Fukushima et al., 2011*) and we confirmed this using both $P_{yocH}$-venus and $P_{iseA}$-venus reporters (*Figure 7A*). To determine whether this domain is required to respond to changes in D,L-endopeptidase activity, we monitored $P_{iseA}$-venus transcription after over-expression of *lytE*. $P_{iseA}$ transcription was de-repressed in both wild-type cells and the Δ*walK* mutant complemented with full-length *walK*, indicative of a decrease in WalRK signaling

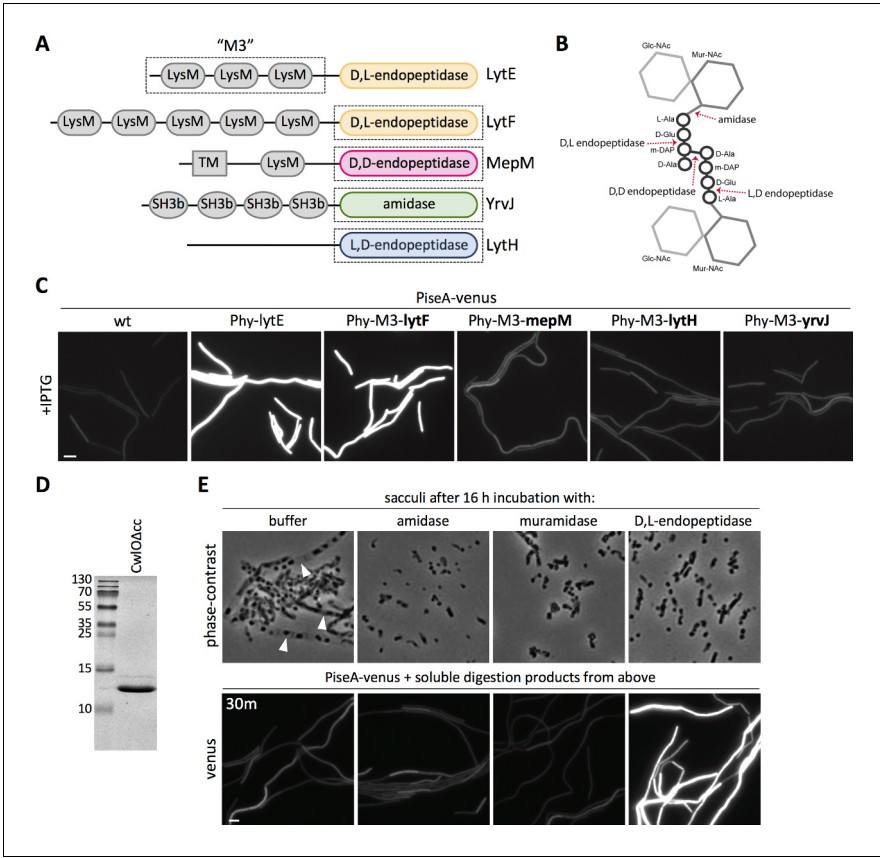

**Figure 8.** The WalRK pathway specifically responds to D,L-endopeptidase-cleaved peptidoglycan. (**A**) Schematic diagram of the cell wall hydrolases used to make LytE chimeras. The three amino-terminal LysM domains of LytE (M3) were fused to the carboxyl-terminal catalytic domains of LytF, MepM, LytH, and YrvJ. (**B**) Schematic of a single PG crosslink and the cleavage sites of the cell wall hydrolase domains used in (**A**); the amino acids in the crossbridge are represented by black circles and the di-saccharide with gray hexagons. (**C**) Representative fluorescence images of the indicated strains harboring P$_{iseA}$-venus reporter and a strong IPTG-inducible promoter (P$_{hy}$) fused to *lytE* or the chimeras. Cells were grown to OD$_{600}$ ~0.3 in LB medium and imaged 30 min after addition of IPTG. 50 µM IPTG was used to induce LytE and the D,L-endopeptidase chimera and 500 µM IPTG was used for all the other chimeras. (**D**) Coomassie-stained gel of purified CwlO lacking its coiled-coil domain (CwlOΔcc). (**E**) Phase-contrast images of purified *B. subtilis* sacculi incubated overnight at 37°C with buffer or purified CwlOΔcc, the amidase LytA, or the muramidase mutanolysin (top panels). Translucent sacculi (carets) surrounding phase-dark insoluble aggregates are only visible after incubation with buffer. Representative fluorescence images of *B. subtilis* cells harboring P$_{iseA}$-venus after 30 min incubation with the soluble material generated from overnight incubation of sacculi under the indicated conditions (bottom panels). Representative images are from one of three independent experiments. Scale bar indicates 5 µm.

The online version of this article includes the following figure supplement(s) for figure 8:

**Figure supplement 1.** D,L-endopeptidase chimeras and the MepM D,D-endopeptidase chimera complement a ΔlytE ΔcwlO double mutant but only D,L-endopeptidase activity lowers WalRK activity.

**Figure supplement 2.** The soluble material after incubating PG sacculi with iodoacetamide-treated CwlO does not lower WalRK activity.

---

(*Figure 7B*). However, the Δ*walK* mutant harboring the *walK*(ΔsCache) variant was unresponsive to *lytE* over-expression (*Figure 7B*). Furthermore, the *walK*(ΔsCache) variant failed to respond to depletion of CwlO (*Figure 7—figure supplement 1*).

These data are consistent with the idea that the sCache domain of WalK is necessary for sensing changes in D,L-endopeptidase activity. However, previous in vivo formaldehyde crosslinking experiments suggested that the interaction between WalK and WalH/WalI requires the extracellular domain of WalK (*Fukushima et al., 2011*). If correct, the constitutive signaling in the *walK*(ΔsCache)

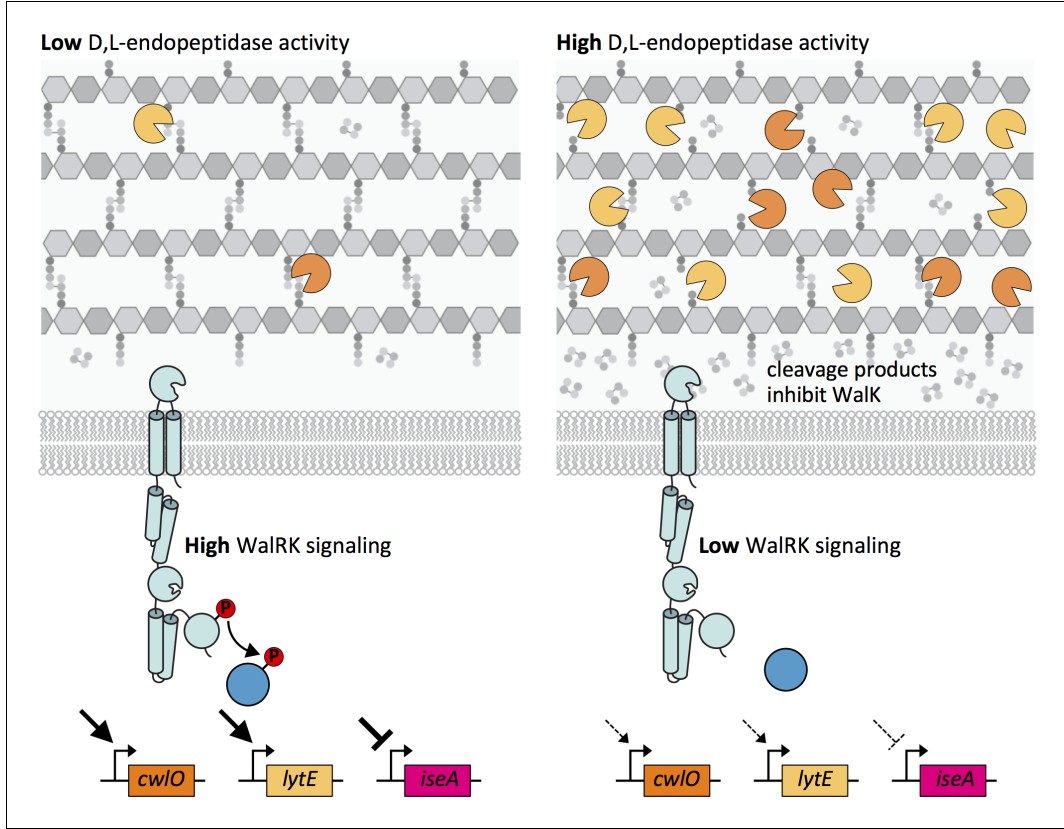

**Figure 9.** Model depicting the homeostatic control of cell wall elongation hydrolases by the WalRK two-component system. The WalK sensor kinase and its two regulators WalH and WalI (not shown) sense and respond to the extent of D,L-endopeptidase cleavage. (Left panel) When the CwlO and LytE activities (orange and yellow scissors) are low, the concentration of their cleavage products is low and WalK kinase activity is high generating high levels of phosphorylated WalR. WalR ~P activates transcription of *lytE* and *cwlO* and represses transcription of *iseA*, encoding a secreted inhibitor of LytE. Thus, increased WalRK signaling returns D,L-endopeptidase activity to its homeostatic set point (not shown). (Right panel) When D,L-endopeptidase activity is high, the concentration of cleavage products generated by these enzymes increases and inhibits WalK signaling leading to a reduction in WalR ~P. Low WalR ~P reduces transcription of *lytE* and *cwlO* and de-represses *iseA* expression. CwlO levels are rapidly reduced due to its short half-life (*Figure 9—figure supplement 1*) and LytE activity is rapidly inhibited by IseA (*Figure 9—figure supplement 2A*). Thus, inhibition of WalK signaling by high concentrations of cleavage products efficiently restores homeostatic levels of the elongation D,L-endopeptidases.

The online version of this article includes the following figure supplement(s) for figure 9:

**Figure supplement 1.** IseA inhibits LytE more potently than the cell separation hydrolases.

**Figure supplement 2.** CwlO has a short half-life during exponential growth.

mutant and the failure to respond to changes in D,L-endopeptidase activity could be due to the loss of inhibition by WalH and WalI. To more directly test the interactions among these proteins, we used the Bacterial Adenylate Cyclase Two Hybrid (BACTH) system (*Ladant and Ullmann, 1999*; *Karimova et al., 2005*). We generated fusions with complementary fragments (T18 and T25) of *Bordetella pertussis* adenylate cyclase to WalK, WalK(ΔsCache), WalH, and WalI. The fusions were co-transformed into *E. coli* and assayed for interaction on LB agar supplemented with X-Gal (see Materials and methods). As can be seen in *Figure 6C*, a strong interaction was only observed when T25-WalK and T18-WalH were co-expressed with untagged WalI, indicating that all three proteins are required for stable interaction. Importantly, we observed a similarly strong interaction using T25-WalK(ΔsCache) (*Figure 6C*). These data argue that the WalK variant lacking its ECD stably interacts with WalH and WalI in vivo. Collectively, these results and those from *Figure 5* suggest that the extracellular sCache domain of WalK functions as the sensing domain that monitors changes in D,L-endopeptidase activity.

## The WalK response is specific to D,L-endopeptidase crosslink cleavage

The data presented thus far indicate that the WalK sensor kinase responds to changes in cell wall hydrolysis. We envisioned two possible models for what WalK might be sensing. Since CwlO and LytE are both D,L-endopeptidases, WalK could specifically monitor D,L-endopeptidase cleavage products. Alternatively, WalK could be sensing some aspect of the cell wall that is affected by PG hydrolysis in general. For example, WalK could monitor the presence of intact crosslinks or the extent of tension on the PG meshwork. To help distinguish between these two models, we investigated whether WalK responds to PG hydrolases that cleave distinct bonds in the PG meshwork (*Figure 8B*). To target these enzymes to the lateral cell wall, we fused the catalytic domains of each PG hydrolase to the N-terminal LysM domains of LytE (M3) (*Figure 8A*) that direct it to the lateral wall (*Buist et al., 2008*; *Hashimoto et al., 2012*; *Hashimoto et al., 2018*). Each chimera was expressed under the control of the same IPTG-regulated promoter used to over-express *lytE*, and WalK signaling was monitored 30 min after induction using the P*iseA*-*venus* reporter. Consistent with the idea that WalK monitors D,L-endopeptidase activity, the chimeras that contained D,L-endopeptidase domains from PG hydrolases that function in cell separation (LytF and CwlS from *B. subtilis*) (*Ohnishi et al., 1999*; *Yamamoto et al., 2003*; *Fukushima et al., 2006*) de-repressed P*iseA* transcription (*Figure 8C*, *Figure 8—figure supplement 1B*). By contrast, WalK signaling was not impacted by expression of chimeras containing D,D-endopeptidase domains (MepM and MepS from *E. coli*) (*Singh et al., 2012*), L,D-endopeptidase domains (LytH and CwlK from *B. subtilis*) (*Horsburgh et al., 2003*; *Fukushima et al., 2007*), the amidase domain from *B. subtilis* YrvJ (*Wendrich and Marahiel, 1997*), or the glucosaminidase domain from *B. subtilis* LytD (*Margot et al., 1994*) (*Figure 8C*, *Figure 8—figure supplement 1B*). Over-expression of most of the chimeras did not cause discernable morphological defects raising the possibility that the fusions were not active enough to alter WalRK signaling, however, we note that the MepM D,D-endopeptidase chimera was able to suppress the lethality of Δ*cwlO* Δ*lytE* double mutant (*Hashimoto et al., 2018*) but even this fusion did not impact P*iseA* transcription (*Figure 8—figure supplement 1A*). Collectively, these data favor the model that WalK specifically monitors D,L-endopeptidase activity.

To directly test whether D,L-endopeptidase cleavage products inhibit WalRK signaling, we sought to investigate whether soluble PG cleavage products generated by a D,L-endopeptidase in vitro could de-repress the P*iseA*-*venus* reporter. To this end, we incubated purified *B. subtilis* sacculi at 37° C for 16 hr with a commercial muramidase (mutanolysin from *Streptomyces globisporus*), a purified amidase (LytA from *Streptococcus pneumoniae*) (*Flores-Kim et al., 2019*), or the constitutively active D,L-endopeptidase domain from *B. subtilis* CwlO that lacks its regulatory coiled-coil domain (CwlOΔcc) (*Figure 8D*). Phase-contrast microscopy was used to assess cell wall hydrolysis (*Figure 8E* - top panels). Translucent sacculi containing phase-dark aggregates (*Liechti et al., 2014*) were readily detectable in buffer-treated sacculi, while only the liberated phase-dark aggregates were present in the sacculi incubated with the PG hydrolases. The soluble material was collected from each digestion and added to cells harboring the P*iseA*-*venus* reporter. Notably, only the cleavage products generated by D,L-endopeptidase digestion reduced WalRK activity and de-repressed P*iseA*-*venus* (*Figure 8E* - bottom panels). Furthermore, the soluble material from sacculi incubated with iodoacetamide-inactivated CwlOΔcc failed to reduce WalRK activity (*Figure 8—figure supplement 2*). These results indicate that WalK specifically responds to D,L-endopeptidase cleavage products.

## Discussion

Altogether, our data support a model in which the WalR-WalK two-component signaling pathway functions in homeostatic control of the cell wall elongation hydrolases LytE and CwlO, and does so by sensing and responding to PG cleavage products generated by these enzymes (*Figure 9*). When D,L-endopeptidase activity is low, the concentration of these cleavage products drops leading to an increase in WalK kinase activity and a concomitant increase in WalR ~P. WalR ~P increases transcription of *lytE* and *cwlO*, restoring homeostasis (*Figure 9* – left panel). Reciprocally, when there is elevated D,L-endopeptidase activity, the concentration of cleavage products increases. High concentrations of these putative allosteric inhibitors reduce WalK activity, resulting in a decrease in *lytE* and *cwlO* transcription (*Figure 9* – right panel).

In this model, transcriptional activation of *lytE* and *cwlO* rapidly boosts D,L-endopeptidase activity when the levels of these enzymes drop. However, this pathway can also rapidly reduce D,L-

endopeptidase activity when it gets too high. The *iseA* gene, which is repressed by WalR ~P, encodes a secreted inhibitor of D,L-endopeptidases (*Bisicchia et al., 2007*; *Salzberg and Helmann, 2007*; *Yamamoto et al., 2008*). It was given its name (inhibitor of cell separation A) because of its ability to inhibit cell separation D,L-endopeptidases when over-expressed on a multi-copy plasmid (*Yamamoto et al., 2008*). However, we have found that LytE is likely to be the primary target of IseA. Specifically, we found that the levels of IseA needed to inhibit LytE had no impact on cell separation (*Figure 9—figure supplement 1A and B*). Furthermore, we have also found that CwlO has a half-life of ~7 min (*Figure 9—figure supplement 2A*). Accordingly, when D,L-endopeptidase activity is too high, a drop in WalRK signaling not only causes a reduction in *lytE* and *cwlO* transcription, but also inhibition of LytE activity via de-repression of *iseA* and a reduction in CwlO levels through degradation or shedding of the enzyme into the medium (*Figure 9—figure supplement 2B*). Thus, the WalRK signaling pathway is able to adjust the levels and activity of CwlO and LytE on the time-scale of a cell cycle to maintain homeostatic control of these essential enzymes during growth.

Previous work on the WalRK signaling pathway led to the model that WalK localization to the septal ring, in the absence of its negative regulators WalH and WalI, couples its activity to growth via the divisome (*Fukushima et al., 2008*; *Dubrac et al., 2008a*). Support for this model comes from experiments in which depletion of FtsZ or other divisome components results in a decrease in *yocH* mRNA levels and an increase in *pdaC* mRNA levels, suggesting that WalK-dependent phosphorylation of WalR requires an intact divisome (*Fukushima et al., 2008*; *Fukushima et al., 2011*). One caveat of these experiments is that *yocH* and *pdaC* transcript levels were monitored 3 hr after depletion of the divisome components, raising the possibility that the reduction in WalR activity could be indirect. We note that we did not detect changes in P*iseA*- and P*yocH*-directed Venus fluorescence after inhibiting FtsZ for 70–90 min (*Figure 5*, *Figure 5—figure supplement 2*, *Figure 5—figure supplement 3*). However, RT-PCR is likely to be more sensitive than our fluorescence-based assay. It is therefore possible that WalRK functions in the homeostatic control of the elongation hydrolases as described here, and separately acts at the divisome to boost CwlO and LytE levels during cytokinesis in anticipation of cell wall growth following division. Time-lapse microscopy in microfluidic devices like the mother machine (*Wang et al., 2010*) will enable a direct test of this model in the future.

We have demonstrated that the soluble cleavage products generated by D,L-endopeptidase-digestion can inhibit WalK signaling and could therefore function as allosteric inhibitors of the kinase. Although the specific cleavage product that inhibits WalK is currently unknown, we favor the idea that the cross-linked tetra- and/or penta-peptides liberated by cleavage of peptide crosslinks (*Figures 8B* and *9*) are the WalK ligand because cleavage of PG crossbridges is required for expansion of the PG meshwork during growth (*Bisicchia et al., 2007*; *Meisner et al., 2013*; *Hashimoto et al., 2012*; *Hashimoto et al., 2018*). To generate these products, both γ-D-Glu-mDAP bonds must be cleaved in the peptide crossbridge. Concerted cleavage of identical bonds in peptide crosslinks has been proposed previously (*Wong et al., 2015*), however it has not been rigorously tested. Alternatively, it is also possible that di- or tri-peptides released upon cleavage of un-cross-linked stem peptides or even short glycan strands with attached di-peptides could serve as the signal. Future experiments will be directed at defining the specific cleavage product that is sensed by WalK and ultimately the ligand-binding site in the sensor domain.

As indicated above, our data are most consistent with a model in which WalK is inhibited when it binds its signal and active when un-liganded (*Figure 9*). Specifically, we showed that D,L-endopeptidase cleavage products cause de-repression of P*iseA*, indicative of low levels of WalR ~P and low WalK kinase activity. Reciprocally, we found that depletion of CwlO, and therefore low concentrations of cleavage products, leads to high P*yocH* transcription, indicative of high levels of WalR ~P and high WalK kinase activity. Another, closely related, two-component system in *B. subtilis* is the PhoR-PhoP system, involved in sensing and responding to phosphate limitation (*Seki et al., 1987*; *Botella et al., 2014*). The PhoP response regulator activates genes involved in phosphate scavenging and inhibits the synthesis of the phosphate-rich surface polymers called wall teichoic acid (WTA). Interestingly, work from the Devine lab suggests that the sensor kinase PhoR is active when un-liganded and inhibited when bound by intermediates in the WTA biosynthetic pathway (*Botella et al., 2014*). In this case, the signal is thought to be sensed by the intracellular PAS domain of PhoR. Nonetheless, it is interesting that in both cases these kinases appear to be regulated by allosteric inhibition.

Cell growth requires an intimate balance between cell wall synthesis and its hydrolysis. The homeostatic control pathway we have defined here ensures that exponentially growing cells maintain a defined amount of D,L-endopeptidase activity for cell wall elongation. However, under different growth conditions, for example entry into stationary phase, the cell is likely to require different levels of hydrolase activity. We propose that the cell could adapt by modulating the homeostatic 'set point' of the WalRK signaling pathway. This set point is likely determined by the levels of the WalK-WalH-WalI sensor kinase complex; the affinity of WalK for its putative allosteric inhibitor; and the activities of the histidine kinase and response regulator. A link between one or more of these determinants and PG synthesis could ensure coordination. Since the *wal* operon is not under auto-regulatory control (*Fabret and Hoch, 1998*), altering the set point is unlikely to occur through changes in the levels in the WalK-WalH-WalI complex. Instead, *B. subtilis* could modulate WalK's affinity for D,L-endopeptidase cleavage products or adjust WalK kinase activity or WalR-dependent transcription. For example, the conserved extracellular domains of WalH and WalI could function in modulating ligand affinity in response to an extracellular signal generated during PG synthesis, or in response to changes in the cell envelope itself. Similarly, the conserved intracellular PAS domain of WalK could alter kinase activity in response to a cytoplasmic signal. For example, this domain could sense intermediates in the PG precursor (lipid II) pathway, analogous to the signal sensed by the PhoR PAS domain. Finally, WalR activity is known to be modulated by PrkC-dependent phosphorylation during entry into stationary phase (*Libby et al., 2015*), which is thought to respond to extracellular muropeptides. By changing the homeostatic set point for hydrolase activity, extracellular signals could directly coordinate hydrolase activity with PG synthesis while the intracellular signal could link hydrolase activity to flux through the precursor synthesis pathway. Identifying these potential allosteric modulators and defining how they impact the homeostatic control of LytE and CwlO are important challenges for the future.

The WalRK two-component system is the most broadly conserved TCS in the Firmicutes and can be found in important human pathogens including *Staphylococcus aureus*, *Listeria monocytogenes*, *Bacillus anthracis*, *Streptococcus pneumoniae*, and *Streptococcus mutans* (*Dubrac et al., 2008a*). In most cases, this pathway is essential and, in the organisms in which it has been investigated, the WalR regulon contains cell wall hydrolases although not necessarily D,L-endopeptidases (*Martin et al., 1999*; *Howell et al., 2003*; *Dubrac and Msadek, 2004*; *Ng et al., 2005*; *Liu, 2006*; *Bisicchia et al., 2007*; *Ahn and Burne, 2007*; *Delaune et al., 2011*). Furthermore, as is the case with *B. subtilis* (*Takada et al., 2018*), the essentiality of this pathway can be bypassed by engineering these bacteria to express a subset of the PG hydrolytic enzymes in the WalR regulon (*Ng et al., 2003*; *Delaune et al., 2011*). Based on these similarities, we hypothesize that the WalRK pathway functions in homeostatic control of cell wall hydrolysis in these pathogens and uses distinct cleavage products to monitor PG hydrolase activity.

It is noteworthy that the WalK sensor kinases in *Streptococci* and *Lactococci* lack an extracellular sCache domain (*Lange et al., 1999*; *Dubrac et al., 2008a*; *Wang et al., 2013*). Accordingly, if there is homeostatic control of cell wall hydrolases in these bacteria they must use a different sensing mechanism or employ a distinct signaling pathway. Intriguingly, the StkP serine/threonine kinase in *S. pneumoniae* has been implicated in modulating cell wall hydrolysis and has been suggested to function in concert with WalRK pathway in this organism (*Fleurie et al., 2012*; *Stamsås et al., 2017*). Analysis of the WalK signaling in diverse Gram-positive pathogens will establish the similarities and differences in the regulatory logic we have uncovered here. Finally, the cleavage products of the PG hydrolases in the WalR regulons could potently inhibit WalK signaling and therefore offer the potential for therapeutic development.

## Materials and methods

### Strains, plasmids, and routine growth conditions

All *Bacillus subtilis* strains were derived from the prototrophic strain PY79 (*Youngman et al., 1983*). Cells were grown in either Luria-Bertani (LB) or casein hydrolysate (CH) medium at 37°C. Unless otherwise indicated, *B. subtilis* strains were constructed using genomic DNA and a 1-step competence method. Antibiotic concentrations were used at: 100 µg/mL spectinomycin, 5 µg/mL chloramphenicol, 10 µg/mL tetracycline, 10 µg/mL kanamycin, 1 µg/mL erythromycin and 25 µg/mL lincomycin. A

list of strains and plasmids used in this study can be found in the Key Resources Table (*Supplementary file 2*), and oligonucleotide primers can be found in *Supplementary file 1*.

## β-Galactosidase assays

*B. subtilis* strains were grown in LB medium at 37°C to an $OD_{600}$ of ~0.5. The optical density was recorded and 1 mL of culture was harvested and assayed for β-galactosidase activity as previously described (*Rudner et al., 1999*). Briefly, cell pellets were re-suspended in 1 mL Z buffer (40 mM $NaH_2PO_4$, 60 mM $Na_2HPO_4$, 1 mM $MgSO_4$, 10 mM KCl, and 50 mM β-mercaptoethanol). 250 μL of this suspension was added to 750 μL of Z buffer supplemented with lysozyme (0.25 mg/ml), and the samples were incubated at 37°C for 15 min. The colorimetric reaction was initiated by addition of 200 μL of 2-nitrophenyl-β-D-galactopyranoside (ONPG, 4 mg/ml) in Z buffer and stopped with 500 μL 1M $Na_2CO_3$. The absorbance at 420 nm and $OD_{550}$ of the reactions were recorded, and the β-galactosidase specific activity in Miller Units was calculated according to the formula $[A_{420}-1.75x(OD_{550})]$ / (time [min] x $OD_{600}$) x dilution factor x 1000 (*Miller, 1972*).

## Immunoblot analysis

Immunoblot analysis was performed as described previously (*Wang et al., 2015*). Briefly, the $OD_{600}$ was recorded for each culture, 1 mL was collected, and the cell pellet re-suspended in lysis buffer (20 mM Tris pH 7.0, 10 mM $MgCl_2$, 1 mM EDTA, 1 mg/mL lysozyme, 10 μg/mL DNase I, 100 μg/mL RNase A, 1 mM PMSF, 1 μg/mL leupeptin, 1 μg/mL pepstatin) to a final $OD_{600}$ of 10 for equivalent loading. The cells were incubated at 37°C for 10 min followed by addition of an equal volume of sodium dodecyl sulfate (SDS) sample buffer (0.25 M Tris pH 6.8, 4% SDS, 20% glycerol, 10 mM EDTA) containing 10% β-mercaptoethanol. Samples were heated for 15 min at 65°C prior to loading. Proteins were separated by SDS-PAGE on 15% (LytE) or 12.5% (FtsX, CwlO, SigA) polyacrylamide gels, electroblotted onto Immobilon-P membranes (Millipore) and blocked in 5% nonfat milk in phosphate-buffered saline (PBS) with 0.5% Tween-20. The blocked membranes were probed with anti-LytE (1:10,000), anti-SigA (1:10,000) (*Fujita and Sadaie, 1998*), anti-FtsX (1:10,000) (*Meisner et al., 2013*), or anti-CwlO (1:10,000) (*Meisner et al., 2013*) diluted into 3% BSA in 1x PBS with 0.05% Tween-20. Primary antibodies were detected using horseradish peroxidase-conjugated goat anti-rabbit IgG (BioRad) and the Super Signal chemiluminescence reagent as described by the manufacturer (Pierce). Signal was detected using a FluorChem R System (Protein Simple).

## LytE purification and antibody production

Recombinant LytE lacking its three N-terminal LysM domains (LytEΔlysMx3) was expressed in *E. coli* BL21 (DE3) using the $P_{T7}$-His$_6$-SUMO-lytEΔlysMx3 expression vector (pYB18). Cells were grown in Terrific Broth (*Tartof and Hobbs, 1988*) supplemented with 100 μg/ml ampicillin at 37°C to $OD_{600}$ = 0.3. LytEΔlysMx3 expression was induced for 16 hr at 22°C by addition of 0.5 mM IPTG. After induction, cells were collected by centrifugation at $10,000 \times g$ for 10 min. Cell pellets were resuspended in 15 mL Buffer 1 (20 mM Tris pH 7.5, 300 mM NaCl, 5 mM imidazole, 10% glycerol, 0.1 μM Dithiothreitol) and Complete EDTA-free protease inhibitors (Roche) and lysed via passage through a French press. Cell lysates were clarified by centrifugation at 10,000 x *g* for 10 min at 4°C. Clarified lysates were mixed with 0.5 mL of $Ni^{2+}$-NTA agarose resin (Qiagen) and incubated for 2 hr at 4°C. The mixture was loaded onto a column (BioRad) and washed with 10 mL Buffer 1. The His$_6$-SUMO-LytEΔlysMx3 fusion protein was eluted with Buffer 2 (20 mM Tris pH 7.5, 300 mM NaCl, 200 mM imidazole, 0.1 μM Dithiothreitol). Eluates were pooled and dialyzed into storage buffer (20 mM Tris pH 7.5, 300 mM NaCl, 10% glycerol, 0.1 μM Dithiothreitol) at 4°C overnight. 10 μL of purified His$_6$-Ulp1 (1.25 mg/ml) was added to the dialysate and was incubated overnight on ice. The reaction was then mixed with 0.5 mL $Ni^{2+}$-NTA agarose and loaded onto a column. Flow-through fractions containing the cleaved (untagged) LytEΔlysMx3 were collected and used to generate rabbit polyclonal antibodies (Covance).

## Fluorescence microscopy

Exponentially growing cells were harvested and concentrated by centrifugation at 6800 x *g* for 1.5 min and re-suspended in 1/10th volume growth medium and then immobilized on 2% (wt/vol) agarose pads containing growth medium. Fluorescence microscopy was performed on a Nikon Ti

inverted microscope equipped with a Plan Apo 100x/1.4 Oil Ph3 DM phase contrast objective, an Andor Zyla 4.2 Plus sCMOS camera, and Lumencore SpectraX LED Illumination. Images were acquired using Nikon Elements 4.3 acquisition software. The fluorescent membrane dye TMA-DPH was added to the concentrated cell suspension at 50 μM final. Propidium iodide (PI) was added at a final concentration of 5 μM. Venus and YFP were imaged using a Chroma ET filter cube for YFP (49003) with an exposure time of 800 ms; TMA-DPH was visualized using a Chroma ET filter cube for DAPI (49000) with an exposure time of 300 ms; mCherry and PI were visualized using a Chroma ET filter cube for mCherry (49008) with an exposure time of 800 ms and 500 ms, respectively. Image processing was performed using Metamorph software (version 7.7.0.0) and Oufti (*Paintdakhi et al., 2016*) was used for quantitative image analysis.

## Bacterial two-hybrid assay

The Bacterial Adenylate Cyclase-based Two Hybrid (BACTH) system was used as previously described (*Karimova et al., 1998*; *Bendezú et al., 2009*). Briefly, pairs of proteins were fused to the complementary fragments (T18 and T25) of the *Bordetella pertusis* adenylate cyclase. Competent BTH101 *E. coli* cells were co-transformed with the two plasmids containing T18 and T25 protein fusions in one step. Transformants were selected on LB agar plates supplemented with 100 μg/mL ampicillin ($Amp^{100}$), 50 μg/mL kanamycin ($Kan^{50}$), 500 μg/mL isopropyl-β-D-thiogalactoside ($IPTG^{500}$), and 100 μg/mL 5-Bromo-4-chloro-3-indolyl-β-D-galactopyranoside ($X\text{-}Gal^{100}$). Plates were incubated at 30°C and the homogeneity of the colony color among transformants was confirmed. Single colonies were then grown to early stationary phase in LB medium supplemented with $Amp^{100}$ and $Kan^{50}$, normalized to $OD_{600} = 0.2$, and spotted (3 μL) on LB agar plates containing $Amp^{100}$, $Kan^{50}$, $IPTG^{500}$, $X\text{-}Gal^{100}$. Plates were incubated at 30°C overnight and imaged. Transformations were done in triplicate with selected images representative of three biological replicates.

## PG hydrolase over-expression

Cultures of exponentially growing cells were diluted to $OD_{600} = 0.01$ in LB and grown for ~1 hr at 37°C to an $OD_{600}$ ~0.05. The cultures were then induced with IPTG (concentrations indicated in the Figure Legends). Fluorescent images were acquired before and at indicated times after induction. For experiments in which cell division was inhibited, cultures of exponentially growing cells were diluted to $OD_{600} = 0.04$ and 10 mM xylose was added to induce *mciZ*. The cultures were then grown for ~1 hr at 37°C to allow cells to filament for ~2 mass doublings before 50 μM IPTG was added to induce *lytE* expression.

## PG hydrolase depletion

Cultures were grown in LB supplemented with 1 mM IPTG to mid-exponential phase, washed three times in LB lacking inducer, and diluted to $OD_{600} = 0.05$ in LB to initiate depletion. Images were acquired before and at indicated times after depletion as described in the text. For experiments in which cell division was inhibited, cultures were grown in the presence of 1 mM IPTG (to induce *cwlO*) until mid-exponential phase. Cells were diluted to $OD_{600} = 0.04$ with 1 mM IPTG and 10 mM xylose (to induce *mciZ*). The cultures were then grown for ~1 hr at 37°C to allow cells to filament for ~2 mass doublings. The cells were then washed three times in LB lacking IPTG, and resuspended in LB containing 10 mM xylose. *iseA* over-expression.

Cultures of exponentially growing cells were diluted to $OD_{600} = 0.02$ in CH medium supplemented with 10 mM xylose and grown for ~1.5 hr at 37°C to an $OD_{600}$ ~0.2. The cultures were then induced with 500 μM IPTG. Fluorescent images were acquired at indicated times after induction.

## In vivo protein turnover assay

Wild-type *B. subtilis* was grown in LB medium at 37°C to an $OD_{600}$ of 0.5. Protein translation was blocked by the addition of both spectinomycin (200 μg/mL, final concentration) and chloramphenicol (10 μg/mL, final concentration). Samples (1 mL of culture) were collected immediately prior to antibiotic treatment and at the indicated times after. Cells were pelleted by centrifugation for 5 min and immediately flash-frozen in liquid nitrogen. The cell pellets were thawed on ice, resuspended in lysis buffer (20 mM Tris pH 7.0, 10 mM $MgCl_2$, 1 mM EDTA, 1 mg/ml lysozyme, 10 μg/ml DNase I, 100 μg/ml RNase A, 1 mM PMSF, 1 μg/ml leupeptin, 1 μg/ml pepstatin), and the suspensions transferred

to fresh microfuge tubes to avoid CwlO present in culture medium that non-specifically bound to the plastic tube (*Brunet et al., 2019*). The lysates were then analyzed by immunoblot as described above.

## CwlOΔcc purification

Recombinant CwlO lacking its N-terminal coiled coil domain (Δcc) was expressed in *E. coli* BL21 (DE3) Δ*fhuA* (New England Biolabs) using the $P_{T7}$-His$_6$-SUMO-cwlOΔcc expression vector (pJM63). Cells were grown in LB supplemented with 100 µg/mL ampicillin at 37°C to $OD_{600}$ = 0.5. Cultures were allowed to equilibrate at room temperature for 30 min and then transferred to 30°C. His$_6$-SUMO-CwlOΔcc expression was induced with 0.5 mM IPTG for 3 hr. Cells were collected by centrifugation, resuspended in 50 mL Buffer A (20 mM Tris HCl pH 7.4, 500 mM NaCl, 20 mM Imidazole, and 2X complete protease inhibitor tablets (Roche), and stored at −80 °C. The cell suspension was thawed on ice and lysed by two passes through a cell disruptor (Constant Systems Ltd.) at 25,000 psi. The lysate was clarified by ultracentrifugation at 35,000 rpm for 30 min at 4°C. The supernatant was added to 1 mL Ni$^{2+}$-NTA beads (Qiagen) and incubated for 1 hr at 4°C. The suspension was loaded into a 10 mL column (BioRad), washed twice with 4 mL Buffer A, and eluted with 2.5 mL Buffer B (20 mM Tris HCl pH 7.4, 500 mM NaCl, 300 mM Imidazole). 10 µL of purified His$_6$-Ulp1 (1.25 mg/ml) was added to the eluate, and the mixture was dialyzed into storage buffer (20 mM Tris HCl pH 8, 100 mM NaCl, 10% glycerol) overnight at 4°C. The next morning 10 µL more His$_6$-Ulp1 was added to the dialysate and incubated for 1 hr at 30°C. The dialysate was mixed with 1 mL of Ni$^{2+}$-NTA beads for 1 hr at 4°C to remove free His$_6$-Ulp1 and His$_6$-SUMO. The suspension was loaded onto a column and the CwlOΔcc-containing flow-through was collected, aliquoted, and stored at −80°C.

## *B. subtilis* sacculi preparation

Wild-type *B. subtilis* was grown in 500 mL LB at 37°C to an $OD_{600}$ = 0.5. Cells were pelleted, re-suspended in 10 mL 0.1 M Tris HCl pH 7.5 with 2% SDS (wt/vol), and boiled for 1 hr. The sample was cooled to room temperature and incubated with Proteinase K Solution (Invitrogen) at a final concentration of 0.4 mg/mL at 50°C for 1 hr. Sacculi were pelleted at 20,000 x g and washed five times with 10 mL ddH$_2$O until free of SDS. The sacculi were then subjected to acid hydrolysis by suspension in 10 mL 1 M HCl at 37°C for 4 hr. Sacculi were then pelleted and washed five times with 10 mL ddH$_2$O. Sacculi were distributed into 10 × 1 mL aliquots and stored at −80°C.

## Sacculi digestion

*B. subtilis* sacculi aliquots were resuspended in 1.4 mL cleavage buffer (25 mM MES pH 5.5) and dispersed in a water-bath sonicator for 30 min. Purified CwlOΔcc (final concentration 0.13 mg/mL), LytA (*Flores-Kim et al., 2019*), final concentration 0.17 mg/mL), and mutanolysin (Sigma Aldrich, 50 units final) were separately added to sonicated sacculi and incubated overnight at 37°C. After the overnight incubation, insoluble material was pelleted at 20,000 g for 15 min and the soluble cleavage products were collected, lyophilized, heat-inactivated (100°C for 20 min), and stored at −80°C. Immediately prior to use, the lypholized material was resuspended in 50 µL ddH$_2$O. When indicated, CwlOΔcc was inactivated with 20 mM iodoacetamide at room temperature in darkness for 30 min prior to addition to sacculi and overnight incubation.

## Assaying soluble cleavage products for inhibition of WalRK signaling

Cultures of exponentially growing cells were diluted to an $OD_{600}$ = 0.01 in LB and grown for 1 hr at 37°C to $OD_{600}$ ~0.05. 50 µL of soluble sacculi cleavage products (prepared as described above) were added to 450 µL of cells and were incubated at 37°C with aeration. Fluorescent images were acquired before and at indicated times after addition.

## Strain and plasmid construction

### Deletion strains

Insertion-deletion mutants were from the *Bacillus* knock-out (BKE) collection (*Koo et al., 2017*) or were generated by isothermal assembly (*Gibson, 2011*) of PCR products followed by direct transformation into *B. subtilis*. All BKE mutants were back-crossed twice into *B. subtilis* PY79. All deletions

were confirmed by PCR. Antibiotic cassette removal was performed using a temperature-sensitive plasmid that constitutively expresses Cre recombinase (*Meeske et al., 2015*).

The following oligonucleotide primers were used to make the indicated strains:

ΔwalRK::erm (oAM475-478); ΔsigI::kan (oYB213/214, oYB215/216); ΔwalHI::tet (oGD137/150, oGD151/152); Δrsgl::spec (oYB178/179, oYB180/181); ΔwalH::erm (BKE collection); Δwall::erm (BKE collection); ΔprkC::erm (BKE collection); Antibiotic cassettes were amplified with (oWX438/439)

## Plasmid construction

### pYB018 [His-SUMO-lytE(ΔlysMx3) (amp)]

pYB018 was generated in a 2-way ligation with an BamHI-XhoI PCR product containing *lytEΔlysMx3* (amplified from PY79 genomic DNA using oligonucleotide primers oYB66 and oYB54) and pTD68 (*Uehara et al., 2010*). The resulting plasmid was sequence-confirmed.

### pYB064 [ycgO::P$_{veg}$-(optRBS)-lacZ (erm)]

pYB064 was generated in an isothermal assembly reaction (*Gibson, 2011*) with a PCR product containing *Pveg-(optRBS)-lacZ* (oligonucleotide primers oYB189 and oYB190 and bMR18 genomic DNA) and pER118 [*ycgO::erm*] cut with BamHI and EcoRI. pER118 is an ectopic integration vector for insertion at *ycgO* (E. Riley and D.Z.R. unpublished). The resulting plasmid was sequence-confirmed. pYB066 [ycgO::P$_{lytE}$-(optRBS)-lacZ (erm)] pYB066 was generated in a 2-way ligation with an EcoRI-HindIII PCR product containing the *lytE* promoter (oligonucleotide primers oYB195 and oYB196 and PY79 genomic DNA) and pYB064. The resulting plasmid was sequence-confirmed.

### pYB069 [amyE::P$_{lytE}$-(optRBS)-lacZ (kan)]

pYB069 was generated in a 2-way ligation with a BamHI-EcoRI fragment containing *PlytE-(optRBS)-lacZ* from pYB066 *and* pER82 [amyE::kan]. pER82 is an ectopic integration vector derived from pDG364 with the *kan* gene replacing the *cat* gene.

### [yvbJ::P$_{hyperspank}$-(optRBS)-lysM$_3$-mepM(D,D-endo) (spec)]

was not able to be propagated in *E. coli* due to toxicity. Instead, the plasmid was cloned by isothermal assembly (as described below) and directly transformed into bYB952 [ΔlytE::kan yvbJ::cat], to generate strain **bYB962 [ΔlytE::kan yvbJ::P$_{hyperspank}$-(optRBS)-lysM$_3$-mepM(D,D-endo) (spec)]**. The resulting construct was sequence-confirmed. The isothermal assembly reaction used to produce [yvbJ::P$_{hyperspank}$-(optRBS)-lysM$_3$-mepM(D,D-endo) (spec)] included (1) a PCR product containing the 5' end of the *lytE* gene encoding the N-terminal LysM domains (oligonucleotide primers oYB245 and oYB246 and PY79 genomic DNA); (2) a PCR product containing the 3' end of the *mepM* gene encoding the D,D-endopeptidase domain (oligonucleotide primers oYB247 and oYB248 and *E. coli* MG1655 genomic DNA) and (3) pMS052 [yvbJ::P$_{hyperspank}$ (spec)] cut by SpeI and SphI. pMS052 is an ectopic integration vector containing the P$_{hyperspank}$ promoter for insertions at the *yvbJ* locus (M. Stanley and D.Z.R., unpublished).

### pYB083 [yhdG::P$_{hyperspank}$-(optRBS)-lytE (cat)]

pYB083 was generated in a 2-way isothermal assembly reaction with a PCR product containing the *lytE* gene with an optimized RBS (oligonucleotide primers oYB245 and oJM104 and PY79 genomic DNA) and pMS018 [yhdG::P$_{hyperspank}$ (cat)] cut by SpeI and SphI. pMS018 is an ectopic integration vector containing the P$_{hyperspank}$ promoter for insertions at the *yvbJ* locus (M. Stanley and D.Z.R., unpublished). The resulting plasmid was sequence-confirmed.

### pYB087 [yvbJ::P$_{hyperspank}$-(optRBS)-lytE (spec)]

pYB087 was generated in a 2-way isothermal assembly reaction with a PCR product containing the *lytE* with an optimized RBS (oligonucleotide primers oYB245 and oJM104 and PY79 genomic DNA) and pMS052 [yvbJ::P$_{hyperspank}$ (spec)] cut by SpeI and SphI. The resulting plasmid was sequence-confirmed.

### pYB095 [yvbJ::P$_{hyperspank}$-(optRBS)-(lysM)$_3$ (cat)]

pYB095 was generated in a 2-way isothermal assembly reaction with a PCR product containing the 5′ end of the *lytE* gene encoding the N-terminal LysM domains (oligonucleotide primers oYB245 and oYB257 and PY79 genomic DNA) and pYB092 [yvbJ::P$_{hyperspank}$ (cat)] cut with NheI and SpeI. pYB092 is an ectopic integration vector containing the P$_{hyperspank}$ promoter for insertions at the *yvbJ* locus. (Y.B and D.Z.R., unpublished). The resulting plasmid was sequence-confirmed.

### pYB097 [yvbJ::P$_{hyperspank}$-(optRBS)-(lysM)$_3$-yrvJ(amidase) (cat)]

pYB097 was generated in a 2-way isothermal assembly reaction with a PCR product containing the 3′ end of the *yrvJ* gene encoding the amidase domain (oligonucleotide primers oYB260 and oYB261 and PY79 genomic DNA) and pYB095 [yvbJ::P$_{hyperspank}$-(optRBS)-(lysM)$_3$ (cat)] cut by NheI. The resulting plasmid was sequence-confirmed.

### pYB098 [yvbJ::P$_{hyperspank}$-(optRBS)-(lysM)$_3$-lytH(L,D-endo) (cat)]

pYB098 was generated in a 2-way isothermal assembly reaction with a PCR product containing 3′ end of the *lytH* gene encoding the L,D-endopeptidase domain (oligonucleotide primers oYB262 and oYB263 and PY79 genomic DNA) and pYB095 cut with NheI.

### pYB099 [yvbJ::P$_{hyperspank}$-(optRBS)-(lysM)$_3$-cwlK(L,D-endo) (cat)]

pYB099 was generated in a 2-way isothermal assembly reaction with a PCR product containing 3′ end of the *cwlK* gene encoding the L,D-endopeptidase domain (oligonucleotide primers oYB264 and oYB265 and PY79 genomic DNA) and pYB095 cut with NheI.

### pYB100 [yvbJ::P$_{hyperspank}$-(optRBS)-(lysM)$_3$-lytF(D,L-endo) (cat)]

pYB100 was generated in a 2-way isothermal assembly reaction with a PCR product containing 3′ end of the *lytF* gene encoding the D,L-endopeptidase domain (oligonucleotide primers oYB266 and oYB267 and PY79 genomic DNA) and pYB095 cut with NheI.

### pYB101 [yvbJ::P$_{hyperspank}$-(optRBS)-(lysM)$_3$-cwlS(D,L-endo) (cat)]

pYB101 was generated in a 2-way isothermal assembly reaction with a PCR product containing 3′ end of the *cwlS* gene encoding the D,L-endopeptidase domain (oligonucleotide primers oYB268 and oYB269 and PY79 genomic DNA) and pYB095 cut with NheI.

### pYB102 [yvbJ::P$_{hyperspank}$-(optRBS)-(lysM)$_3$-mepS(D,D-endo) (cat)]

pYB102 was generated in a 2-way isothermal assembly reaction with a PCR product containing 3′ end of the *mepS* gene encoding the D,D-endopeptidase domain (oligonucleotide primers oYB270 and oYB271 and *E. coli* MG1655 genomic DNA) and pYB095 cut with NheI.

### pYB103 [yvbJ::P$_{hyperspank}$-(optRBS)-(lysM)$_3$-lytD(glucosaminidase) (cat)]

pYB103 was generated in a 2-way isothermal assembly reaction with a PCR product containing 3′ end of the *lytD* gene encoding the glucosaminidase domain (oligonucleotide primers oYB272 and oYB273 and PY79 genomic DNA) and pYB095 cut with NheI.

### pYB114 [yvbJ::P$_{spank}$-(optRBS)-lytE (spec)]

pYB114 was generated in a 2-way isothermal assembly reaction with a PCR product containing the lytE gene and optimized RBS (oligonucleotide primers oYB245 and oJM104 from PY79 genomic DNA) and pMS050 [yvbJ::P$_{spank}$ (spec)] cut by SpeI and SphI. pMS050 is an ectopic integration vector containing the P$_{spank}$ promoter for insertion at the *yvbJ* locus (M. Stanley and D.Z.R., unpublished). The resulting plasmid was sequence-confirmed.

### pYB139 [ycgO::P$_{yocH}$-(optRBS)-lacZ (erm)]

pYB139 was generated in a 2-way ligation with an EcoRI-HindIII fragment containing the *yocH* promoter from pGD055 [amyE::P$_{yocH}$-(optRBS)-venus (cat)] and pYB064 [ycgO::P$_{veg}$-(optRBS)-lacZ (erm)] cut with the same enzymes.

### pYB140 [ycgO::P$_{iseA}$-(optRBS)-lacZ (erm)]

pYB140 was generated in a 2-way ligation with an EcoRI-HindIII fragment containing the *iseA* promoter from pGD056 [amyE::PiseA-(optRBS)-venus (cat)] and pYB064 cut with the same enzymes.

### pYB142 [yhdG::P$_{xylA}$-(nativeRBS)-lytE (kan])

pYB142 was generated in a 2-way ligation with an EcoRI-BamHI fragment containing *P$_{xylA}$-(nativeRBS)-lytE* from pYB061 [yvbJ:: P$_{xylA}$-(nativeRBS)-lytE (cat)] (Brunet et al., 2019) and pCB059 [yhdG::Pspank (kan)] cut with the same enzymes. pCB059 is an ectopic integration vector for insertions at the *yhdG* locus (R. Barajas and D.Z.R., unpublished).

### pYB159 [yvbJ::P$_{hyperspank}$-(optRBS)-lytE(C247S) (spec)]

pYB159 was generated in a 3-way isothermal assembly reaction. Two of the fragments contained the 5' and 3' regions of the *lytE* gene PCR amplified from *B. subtilis* PY79 genomic DNA using oYB245 and oGD165 and oGD166 and oJM104. Assembly of these two products generated the C247S mutation. The third fragment was pMS052 [yvbJ::Phyperspank (spec)] cut by SpeI and SphI. The resulting plasmid was sequence-confirmed.

### pYB161 [yvbJ::lytE(WalR box2 mut) (spec)]

pYB161 was generated in a 3-way isothermal assembly reaction. Two of the fragments contained the 5' and 3' regions of the *lytE* gene PCR amplified from *B. subtilis* PY79 genomic DNA using oligonucleotide primers oYB352 and oYB336, and oYB337 and oYB353. Assembly of these two products generated the WalR binding site box 2 mutations (WalR box2 mut). The third fragment was pCB043 [yvbJ::spec] cut with EcoRI and BamHI. pCB043 is an ectopic integration vector for insertions at the *yvbJ* locus (R. Barajas and D.Z.R., unpublished). The resulting plasmid was sequence-confirmed.

### pYB162 [yvbJ::lytE (spec)]

pYB162 was generated in a 2-way isothermal assembly reaction with a PCR product containing the *lytE* gene (amplified from PY79 genomic DNA with oligonucleotide primers oYB352 and oYB353) and pCB043 [yvbJ::spec] cut with EcoRI and BamHI. The resulting plasmid was sequence-confirmed.

### pYB164 [ycgO::P$_{lytE(walR \ box2 \ mut)}$-(optRBS)-lacZ (erm)]

pYB164 was generated in a 2-way ligation with an EcoRI-HindIII PCR product containing *P$_{lytE(WalR box2-mut)}$* (amplified from pYB161 [yvbJ::lytE(WalR box2 mut) (spec)] with oligonucleotide primers oYB195 and oYB196) and pYB064 [ycgO::Pveg-(optRBS)-lacZ (erm)] cut with the same enzymes. The resulting plasmid was sequence-confirmed. pYB169 [yhdG::P$_{xylA}$-(optRBS)-walI (erm)] pYB169 was generated in a 2-way isothermal assembly reaction with a PCR product containing the *walI* gene with optimized RBS (amplified from PY79 genomic DNA with oligonucleotide primers oYB351 and oYB361) and pCB106 [yhdG::P$_{xylA}$ (erm)] cut with HindIII and XhoI. pCB106 is an ectopic integration vector containing the P$_{xylA}$ promoter for insertions at the *yhdG* locus (R. Barajas and D.Z.R., unpublished). The resulting plasmid was sequence-confirmed.

### pYB170 [yhdG::P$_{xylA}$-(optRBS)-walIΔECD(36-280) (erm)]

pYB170 was generated in a 2-way isothermal assembly reaction with a PCR product containing *walI*-ΔECD(36-280) (amplified from PY79 genomic DNA with oligonucleotide primers oYB359 and oYB361) and pCB106 [yhdG::P$_{xylA}$ (erm)] cut with HindIII and XhoI. The resulting plasmid was sequence-confirmed.

### pYB177 [amyE::P$_{lytE}$-(optRBS)-venus (cat)]

pYB177 was generated in 2-way isothermal assembly reaction with a PCR product containing the *lytE* promoter (amplified from PY79 genomic DNA with oligonucleotide primers oYB376 and oYB377) and pGD015 [amyE::(optRBS)-venus (cat)] cut with EcoRI. The resulting plasmid was sequence-confirmed.

### pYB190 [ycgO::P$_{hyperspank}$-(optRBS)-iseA (erm)]

pYB190 was constructed in a 2-way ligation with a PCR product containing *iseA* (amplified from PY79 genomic DNA with oligonucleotide primers oYB398 and oYB399) cut with SpeI and HindIII, and pCB089 [ycgO::P$_{hyperspank}$ (erm)] cut with the same enzymes. pCB089 is an ectopic integration vector containing the P$_{hyperspank}$ promoter for insertions at the *ycgO* locus (R. Barajas and D.Z.R., unpublished). The resulting plasmid was sequence-confirmed.

### pAM187 [yvbJ::P$_{spank}$-(nativeRBS)-walRK (spec)]

pAM187 was constructed in a 2-way ligation with a PCR product containing the *walR* and *walK* genes (amplified from *B. subtilis* PY79 genomic DNA with oligonucleotide primers oAM473 and oAM474) cut with XmaI and SpeI and pMS050 [yvbJ::P$_{spank}$ (spec)] cut with the same enzymes. The resulting plasmid was sequence-confirmed. pGD015 [amyE::(optRBS)-venus (cat)] pGD015 was constructed in a 2-way isothermal assembly reaction with a PCR product containing the *venus* gene and an optimized RBS (amplified from plasmid pLPT10 (kindly provided by Johan Paulsson) using oligonucleotide primers oGD59 and oGD60) and pDG364 [amyE::cat] cut with EcoRI and BamHI. The resulting plasmid was sequence-confirmed.

### pGD018 [amyE::P$_{iseA}$-(optRBS)-venus (cat)]

pGD018 was created in a 2-way isothermal assembly reaction with a PCR product containing the *iseA* promoter (amplified from *B. subtilis* PY79 genomic DNA using oligonucleotide primers oGD68 and oGD69) and cloned pGD015 [amyE::(optRBS)-venus (cat)] cut with EcoRI. The resulting plasmid was sequence-confirmed.

### pGD021 [yhdG::P$_{xylA}$-spoIVFB(E44Q)-gfp (kan)]

pGD021 was constructed in a 2-way ligation with a HindIII-BamHI fragment containing *spoIVFB* (E44Q)-*gfp* from pKM260 [(ycgO::P$_{IVF}$-spoIVF(E44Q)-gfp (erm)] and pMS033 [yhdG::P$_{xylA}$ (kan)] cut with the same enzymes. pMS033 is an ectopic integration vector with P$_{xylA}$ for insertions at the *yhdG* locus (M. Stanley and D.Z.R., unpublished). The resulting plasmid was sequence confirmed.

### pGD022 [yhdG::P$_{xylA}$-(optRBS)-mciZ (kan)]

pGD022 was constructed in a 2-way ligation with a HindIII-BamHI fragment containing the *mciZ* gene from pRB099 [(yvbJ::P$_{xylA}$-mciZ (erm)] and pMS033 [yhdG::P$_{xylA}$ (kan)] cut with the same enzymes. The resulting plasmid was sequence confirmed.

### pGD055 [amyE::P$_{yocH}$-(optRBS)-venus (cat)]

pGD055 was constructed in a 2-way isothermal assembly reaction with a PCR product containing the *yocH* promoter (amplified from *B. subtilis* PY79 genomic DNA using oligonucleotide primers oGD148 and oGD149) and pGD015 cut with EcoRI. The resulting plasmid was sequenced-confirmed. pGD061 [yhdG::P$_{xylA}$-(optRBS)-mciZ (phleo)] pGD061 was constructed in a 2-way ligation with a HindIII-BamHI fragment containing the *mciZ* gene from pRB099 [(yvbJ::P$_{xylA}$-mciZ (erm)] and pCB109 [yhdG::phleo] cut with the same enzymes. pCB109 is an ectopic integration vector for insertions in the *yhdG* locus (R. Barajas and D.Z.R., unpublished).

### pGD062 [yhdG::walRK (kan)]

pGD062 was constructed in a 2-way isothermal assembly reaction with a PCR product containing *walRK* (amplified from *B. subtilis* PY79 genomic DNA using oligonucleotide primers oGD172 and oGD173) and pCB037 [yhdG::kan] cut with EcoRI and SpeI. pCB037 is an ectopic integration vector for insertions in the *yhdG* locus (R. Barajas and D.Z.R., unpublished). The plasmid was sequenced and used for the construction of pGD073.

### pGD073 [yhdG::walRKHI (kan)]

pGD073 was constructed in a 2-way isothermal assembly reaction using a PCR product containing *walKHI* (amplified from *B. subtilis* PY79 genomic DNA using oligonucleotide primers oGD188 and

oGD189) and pGD062 cut with XhoI. XhoI cuts in the middle of *walK*; this feature was used to insert *walKHI* into pGD062. The resulting plasmid was sequenced-confirmed.

### pGD090 [yhdG::walRK(Δ44–167)-walHI (tet)]

pGD090 was constructed in a 3-way isothermal assembly reaction with a PCR product containing *walR-walK(1-43)* (amplified from *B. subtilis* PY79 genomic DNA using oligonucleotide primers oGD172 and oGD215); a PCR product containing *walK(168-611)-walHI* (amplified from *B. subtilis* PY79 genomic DNA using oligonucleotide primers oGD214 and oGD189) and pCB036 [yhdG::tet] cut with EcoRI and XhoI. pCB036 is an ectopic integration vector for insertions in the *yhdG* locus (R. Barajas and D.Z.R., unpublished). The resulting plasmid was sequenced-confirmed.

### pGD101 [ycgO::P$_{xylA}$-(optRBS)-walH (kan)]

pGD101 was constructed in a 2-way isothermal assembly reaction with a PCR product containing *walH* with an optimized RBS (amplified from *B. subtilis* PY79 genomic DNA using oligonucleotide primers oGD227 and oGD228) and pCB136 [yhdG::P$_{xylA}$ (kan)] cut with XhoI and BamHI. pCB136 is an ectopic integration vector with a P$_{xylA}$ promoter for insertions into the *yhdG* locus (R. Barajas and D. Z.R., unpublished). The resulting plasmid was sequenced-confirmed.

### pGD102 [ycgO::P$_{xylA}$-(optRBS)-walHΔECD(61-455) (kan)]

pGD102 was constructed in a 2-way isothermal assembly reaction with a PCR product containing *walH*ΔECD(61-455) and an optimized RBS (amplified from *B. subtilis* PY79 genomic DNA using oligonucleotide primers oGD227 and oGD229) and pCB136 cut with XhoI and BamHI. The resulting plasmid was sequenced-confirmed.

## Bacterial two-hybrid plasmids

### pGD115 [T25-walK]

pGD115 was constructed in a 2-way ligation with a PCR product containing the *walK* gene (amplified from *B. subtilis* PY79 genomic DNA using oligonucleotide primers oGD278 and oGD279) cut with XbaI and KpnI and pKT25 cut with the same enzymes. The resulting plasmid was sequenced-confirmed.

### pGD118 [T18-walH]

pGD118 was constructed in a 2-way ligation with a PCR product containing the *walH* gene (amplified from *B. subtilis* PY79 genomic DNA using oligonucleotide primers oGD282 and oGD283) cut with XbaI and KpnI and pCH364 cut with the same enzymes. The resulting plasmid was sequenced-confirmed.

### pGD120 [T18-walI]

pGD120 was constructed in a 2-way ligation with a PCR product containing the *walI* gene (amplified from *B. subtilis* PY79 genomic DNA using oligonucleotide primers oGD284 and oGD285) cut with XbaI and KpnI and pCH364 cut with the same enzymes. The resulting plasmid was sequenced-confirmed.

### pGD122 [T18-walHI]

pGD122 was constructed in a 2-way ligation with a PCR product containing the *walHI* genes (amplified from *B. subtilis* PY79 genomic DNA using oligonucleotide primers oGD282 and oGD285) cut with XbaI and KpnI and pCH364 cut with the same enzymes. The resulting plasmid was sequenced-confirmed.

### pGD123 [T25-walK(Δ44–167)]

pGD123 was constructed in a 2-way ligation with a PCR product containing the *walK(Δ44–167)* gene (amplified with genomic DNA from bGD500 using oligonucleotide primers oGD278 and oGD279) cut with XbaI and KpnI and pKT25 cut with the same enzymes. The resulting plasmid was sequenced-confirmed.

### pJM063 [$P_{T7}$-SUMO-6xHis-cwlO$\Delta$cc]

pJM063 was constructed in a 2-way isothermal assembly reaction with a PCR product containing the C-terminal domain of CwlO (cwlO$\Delta$cc, amino acids 334–473) (amplified from *B. subtilis* PY79 genomic DNA using oligonucleotide primers oJM117 and oJM320) and pTB146 (*Bendezú et al., 2009*) cut with SacI and BamHI. The resulting plasmid was sequenced-confirmed.

## Acknowledgements

We thank all members of the Bernhardt-Rudner super-group past and present for helpful advice, discussions, and encouragement; Paula Montero Llopis and the HMS Microscopy Resources on the North Quad (MicRoN) core for advice on microscopy; Jeff Meisner for the CwlO expression plasmid and Chris Sham for enzymatic characterization; Alex Meeske for plasmid construction; Johan Paulsson for plasmids; Hoong Chuin Lim for help with fluorescence quantification; James Hoch and Hendrik Szurmant for advice and reagents; Andrew Kruse and Michael Springer for helpful discussions. Support for this work comes from the National Institute of Health Grants GM086466, GM127399, U19 AI109764 (DZR). YRB was funded in part by an EMBO Long-Term Fellowship. JFK was funded in part National Institutes of Health Grant F32AI36431.

## Additional information

### Funding

| Funder | Grant reference number | Author |
|---|---|---|
| National Institute of General Medical Sciences | GM086466 | David Z Rudner |
| National Institute of General Medical Sciences | GM127399 | David Z Rudner |
| National Institute of Allergy and Infectious Diseases | U19 AI109764 | David Z Rudner |
| National Institute of Allergy and Infectious Diseases | F32AI36431 | Josué Flores-Kim |
| European Molecular Biology Organization | Long-Term Fellowship | Yannick R Brunet |

The funders had no role in study design, data collection and interpretation, or the decision to submit the work for publication.

### Author contributions

Genevieve S Dobihal, Yannick R Brunet, Conceptualization, Formal analysis, Validation, Investigation, Methodology; Josué Flores-Kim, Investigation, Methodology; David Z Rudner, Conceptualization, Supervision, Funding acquisition, Project administration

### Author ORCIDs

Genevieve S Dobihal (iD) https://orcid.org/0000-0001-7589-1133
Josué Flores-Kim (iD) http://orcid.org/0000-0001-8282-6647
David Z Rudner (iD) https://orcid.org/0000-0002-0236-7143

### Decision letter and Author response

Decision letter https://doi.org/10.7554/eLife.52088.sa1
Author response https://doi.org/10.7554/eLife.52088.sa2

## Additional files

### Supplementary files

• Supplementary file 1. Table of oligonucleotides used in this study.

- Supplementary file 2. Key resources table.
- Transparent reporting form

### Data availability

All data generated or analysed during this study are included in the manuscript and supporting files.

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
