## [Decision Letter]

**Acceptance summary:**

In this study, the authors show that in *Bacillus subtilis*, cell wall remodeling acts upstream of a regulatory feedback loop that adjusts cell wall homeostasis on the time scale of the cell cycle. In this process, activation of the WalK kinase occurs when D,L-endopeptidase activity is low, leading to increased expression of endopeptidase-encoding genes. On the contrary, at high D,L-endopeptidase activity, the subsequent accumulation of cell wall cleavage products inhibits the kinase and therefore endopeptidase gene expression is shut down. The uncovered regulatory logic is an important step toward understanding of essential WalRK-dependent regulations in other firmicutes including important human pathogens. Conserved themes but also variations are to be expected.

**Decision letter after peer review:**

Thank you for submitting your article "Homeostatic control of cell wall hydrolysis by the WalRK two-component signaling pathway in *Bacillus subtilis*" for consideration by *eLife*. Your article has been reviewed by two peer reviewers, and the evaluation has been overseen by Tâm Mignot as Reviewing Editor and Gisela Storz as the Senior Editor. The following individual involved in review of your submission has agreed to reveal their identity: Hendrik Szurmant (Reviewer #2).

The reviewers have discussed the reviews with one another and the Reviewing Editor has drafted this decision to help you prepare a revised submission.

The WalRK two-component signal transduction system is a widely studied pathway, due to its requirement for cell viability in a wide range of Gram-positive bacteria. This manuscript demonstrates nicely that the *Bacillus subtilis* version of this system adjust its activity in response to the activity of specific autolysins. In turn the genes for these autolysins are under transcriptional control of the WalRK system, thus providing evidence of a homeostatic feedback control system and adding a significant piece to the puzzle. The data is further suggestive that a peptidoglycan cleavage product serves as an inhibitor of WalK kinase activity.

The reviewers only have editorial concerns that require text adjustments prior to publication:

1) One concern with the current manuscript is a lack of reconciliation of the model in Figure 9 with previous studies that demonstrated that WalK is primarily located at the septum in actively growing *B. subtilis* cells. As the authors point out in the subsection “The WalK sensor kinase responds to changes in D,L-endopeptidase activity in the absence of cell division”, WalK is responsive to autolysin activity when FtsZ is inhibited. Certainly it appears that a previous model based on available data at the time was not entirely correct. In this model it was suggested that WalK is active during rapid growth via its location at the septum and inactive under non-growing conditions while in complex with WalHI at the lateral cell wall. The authors here clearly show that WalHI-complexed WalK at the lateral cell wall can be activated by lack of CwlO or LytE activity. Nonetheless, the data that showed WalK localized to the septum is solid and so are published data that show that WalK is responsive (less active) when various cell division proteins are depleted. Interestingly, a soluble fragment of WalK that excludes the extra-cytoplasmic PAS domain and TM helices of WalK maintained its septum localization. Assays suggested that a cytoplasmic PAS domain was essential for this localization. It thus seems plausible that WalK might have two different sensing roles depending on localization, and perhaps mediated by different input domains. Figure 9 nicely summarizes the authors data and is easy to understand to the non-expert reader and it could thus be left as such. But the authors should include a discussion that integrates their new findings with data from previous studies. Similarly, the text in the subsection “The WalK sensor kinase responds to changes in D,L-endopeptidase activity in the absence of cell division” could be amended to not seem as dismissive of previous results.

2) In the Abstract the authors very definitively state that it is the cleavage products that are sensed by the WalK kinase. In the main text they are a little bit more careful with this statement, which I think is adequate. The statement should also be toned down a bit in the Abstract as well.

3) In the first paragraph of the subsection “LytE levels increase in the absence of CwlO maintaining cell envelope integrity”, authors describe a modest but reproducible effect. Could the authors include a statistical test?

---

## [Author Response]

[…] The reviewers only have editorial concerns that require text adjustments prior to publication:1) One concern with the current manuscript is a lack of reconciliation of the model in Figure 9 with previous studies that demonstrated that WalK is primarily located at the septum in actively growing B. subtilis cells. As the authors point out in the subsection “The WalK sensor kinase responds to changes in D,L-endopeptidase activity in the absence of cell division”, WalK is responsive to autolysin activity when FtsZ is inhibited. Certainly it appears that a previous model based on available data at the time was not entirely correct. In this model it was suggested that WalK is active during rapid growth via its location at the septum and inactive under non-growing conditions while in complex with WalHI at the lateral cell wall. The authors here clearly show that WalHI-complexed WalK at the lateral cell wall can be activated by lack of CwlO or LytE activity. Nonetheless, the data that showed WalK localized to the septum is solid and so are published data that show that WalK is responsive (less active) when various cell division proteins are depleted. Interestingly, a soluble fragment of WalK that excludes the extra-cytoplasmic PAS domain and TM helices of WalK maintained its septum localization. Assays suggested that a cytoplasmic PAS domain was essential for this localization. It thus seems plausible that WalK might have two different sensing roles depending on localization, and perhaps mediated by different input domains. Figure 9 nicely summarizes the authors data and is easy to understand to the non-expert reader and it could thus be left as such. But the authors should include a discussion that integrates their new findings with data from previous studies. Similarly, the text in the subsection “The WalK sensor kinase responds to changes in D,L-endopeptidase activity in the absence of cell division” could be amended to not seem as dismissive of previous results.

We have amended the Results and Discussion presenting previous observation in a more balanced fashion.

The relevant paragraph in the Results section now reads:

“Previous work on the WalRK pathway indicate that WalK, but not WalH or WalI, localizes to the septal ring and does so in an FtsZ-dependent manner (Fukushima et al., 2008). […] To address this discrepancy, we investigated whether the WalRK pathway could respond to changes in D,L-endopeptidase activity in cells lacking a divisome.”

The following paragraph was added to the Discussion section:

“Previous work on the WalRK signaling pathway led to the model that WalK localization to the septal ring, in the absence of its negative regulators WalH and WalI, couples its activity to growth via the divisome (Fukushima et al., 2008, Dubrac et al., 2008). […] Time-lapse microscopy in microfluidic devices like the mother machine will enable a direct test of this model in the future (Wang et al., 2010).”

2) In the Abstract the authors very definitively state that it is the cleavage products that are sensed by the WalK kinase. In the main text they are a little bit more careful with this statement, which I think is adequate. The statement should also be toned down a bit in the Abstract as well.

We have toned down our conclusions in the Abstract. It now reads:

"Here, we provide evidence that the essential and broadly conserved WalR-WalK two component regulatory system continuously monitors changes in the activity of these hydrolases by sensing the cleavage products generated by these enzymes and modulating their levels and activity in response."

3) In the first paragraph of the subsection “LytE levels increase in the absence of CwlO maintaining cell envelope integrity”, authors describe a modest but reproducible effect. Could the authors include a statistical test?

We have run a Welch's unequal variances t-test for the data in Figure 1B and include p-values in the figure legend.